# Evolution of C4 photosynthesis predicted by constraint-based modelling

**Mary-Ann Blätke[1]\*, Andrea Bräutigam[1,2]**

[1]Leibniz Institute of Plant Genetics and Crop Plant Research (IPK), Gatersleben, Germany; [2]Computational Biology, Faculty of Biology, Bielefeld University, Universitätsstraße, Bielefeld, Germany

**Abstract** Constraint-based modelling (CBM) is a powerful tool for the analysis of evolutionary trajectories. Evolution, especially evolution in the distant past, is not easily accessible to laboratory experimentation. Modelling can provide a window into evolutionary processes by allowing the examination of selective pressures which lead to particular optimal solutions in the model. To study the evolution of C4 photosynthesis from a ground state of C3 photosynthesis, we initially construct a C3 model. After duplication into two cells to reflect typical C4 leaf architecture, we allow the model to predict the optimal metabolic solution under various conditions. The model thus identifies resource limitation in conjunction with high photorespiratory flux as a selective pressure relevant to the evolution of C4. It also predicts that light availability and distribution play a role in guiding the evolutionary choice of possible decarboxylation enzymes. The data shows evolutionary CBM in eukaryotes predicts molecular evolution with precision.

**\*For correspondence:**
blaetke@ipk-gatersleben.de

**Competing interests:** The authors declare that no competing interests exist.

## Introduction

Identifying specific evolutionary trajectories and modelling the outcome of adaptive strategies at the molecular levels is a major challenge in evolutionary systems biology (*Papp et al., 2011*). The evolution of novel metabolic pathways from existing parts may be predicted using constraint-based modelling (CBM) (*Orth et al., 2010*). In CBM, selective pressures are coded via the objective functions for which the model is optimised. The factors which constrain evolution are integrated into the models via changes in model inputs or outputs and via flux constraints. We hypothesised that the evolution of the agriculturally important trait of C4 photosynthesis is accessible to CBM.

C4 photosynthesis evolved independently in at least 67 independent origins in the plant kingdom (*Scheben et al., 2017*) and it allows colonisation of marginal habitats (*Sage et al., 2012*) and high biomass production in annuals such as crops (*Sage, 2004*; *Edwards et al., 2010*). The C4 cycle acts as a biochemical pump which enriches the $CO_2$ concentration at the site of Rubisco to overcome a major limitation of carbon fixation (*Sage, 2004*). Enrichment is beneficial because Rubisco, the carbon fixation enzyme, can react productively with $CO_2$ and form two molecules of 3-PGA, but it also reacts with $O_2$ and produces 2-phosphoglycolate which requires detoxification by photorespiration (*Ogren and Bowes, 1971*). The ratio between both reactions is determined by the enzyme specificity towards $CO_2$, by the temperature, and the concentrations of both reactants, which in turn is modulated by stresses such as drought and pathogen load. Evolution of Rubisco itself is constrained since any increase in specificity is paid for by a reduction in speed (*Spreitzer and Salvucci, 2002*). Lower speeds most likely cause maladaptivity since Rubisco is a comparatively slow enzyme and can comprise up to 50% of the total leaf protein (*Ellis, 1979*). In the C4 cycle, phosphoenolpyruvate carboxylase affixes $CO_2$ to a C3 acid, phosphoenolpyruvate (PEP), forming a C4 acid, oxaloacetate (OAA). After stabilisation of the resulting C4 acid by reduction to malate or transamination to aspartate, it is transferred to the site of Rubisco and decarboxylated by one of three possible decarboxylation enzymes, NADP-dependent malic enzyme (NADP-ME), NAD-dependent malic enzyme (NAD-

**eLife digest** Virtually all plants use energy from sunlight to convert carbon dioxide and water into oxygen and sugars via a process called photosynthesis. This process has many steps that each rely on different enzymes to drive specific chemical reactions. Most plants use a pathway of enzymes that is referred to as C3 photosynthesis.

Plants absorb carbon dioxide gas from the atmosphere. However, the levels of carbon dioxide in the atmosphere are very low, so this limits the amount of photosynthesis plants can perform. To overcome this problem, some plants have evolved a different type of photosynthesis – called C4 photosynthesis – with a mechanism that increases the levels of carbon dioxide in the cells.

Today, plants that use C4 photosynthesis (so-called 'C4 plants') typically grow faster than other plants, especially in warmer climates. This gives C4 plants, such as corn, an advantage over their competitors and also helps them to colonize harsh environments that other plants struggle to thrive in. However, it remains unclear how C4 photosynthesis evolved in some plants living in wet habitats, or why other plants use forms of photosynthesis that are intermediate between C4 and C3 photosynthesis.

C4 photosynthesis uses pathways containing enzymes that are found in all plants; therefore, C4 plants evolved by changing how they used enzymes they already had. To understand how these different enzyme pathways may have evolved, Blätke and Bräutigam used an approach known as constraint-based modelling. The researchers built a mathematical model of C3 photosynthesis and used it to predict the optimal enzyme pathways (for example, pathways involving the fewest enzymes or requiring the least energy) for photosynthesis under particular conditions.

The model predicted that, in addition to shortages in carbon dioxide, shortages in an important plant nutrient known as nitrogen may have driven the evolution of C4 photosynthesis. Furthermore, enzyme pathways that were intermediate between C3 and C4 photosynthesis were predicted to be optimal solutions under particular conditions. Together, the findings of Blätke and Bräutigam may explain why different variations of C4 photosynthesis exist in plants. These findings could be used to breed crops that use the most efficient type of photosynthesis for the conditions they are grown in, leading to better yields.

ME), or PEP carboxykinase (PEP-CK) (*Hatch, 1987*; *Schlüter et al., 2016b*). Species such as corn (*Zea mays*) (*Pick et al., 2011*) and great millet (*Sorghum bicolor*) (*Döring et al., 2016*) use NADP-ME, species like common millet (*Panicum miliaceum*) (*Hatch, 1987*) and African spinach (*Gynandropsis gynandra*) (*Feodorova et al., 2010*; *Voznesenskaya et al., 2007*) use NAD-ME and species such as guinea grass (*Panicum maximum*) (*Bräutigam et al., 2014*) use mainly PEP-CK with the evolutionary constraints leading to one or the other enzyme unknown. Mixed forms are only known to occur between a malic enzyme and PEP-CK but not between both malic enzymes (*Wang et al., 2014*). After decarboxylation, the C3 acid diffuses back to the site of phosphoenolpyruvate carboxylase (PEPC) and is recycled for another C4 cycle by pyruvate phosphate dikinase (PPDK) (*Hatch, 1987*; *Schlüter et al., 2016b*). All the enzymes involved in the C4 cycle are also present in C3 plants (*Aubry et al., 2011*). In its most typical form, this C4 cycle is distributed between different cell types in a leaf in an arrangement called Kranz anatomy (*Haberlandt, 1904*). Initial carbon fixation by PEPC occurs in the mesophyll cell, the outer layer of photosynthetic tissue. The secondary fixation by Rubisco after decarboxylation occurs in an inner layer of photosynthetic tissue, the bundle sheath which in turn surrounds the veins. Both cells are connected by plasmodesmata which are pores with limited transfer specificity between cells. A model which may test possible carbon fixation pathways at the molecular level thus requires two cell architectures connected by transport processes (*Bräutigam and Weber, 2010*).

CBM of genome-scale or close to it are well suited to study evolution (summarised in *Papp et al., 2011*). Evolution of different metabolic modes from a ground state, the metabolism of *Escherichia coli*, such as glycerol usage (*Lewis et al., 2010*) or endosymbiotic metabolism (*Pál et al., 2006*) have been successfully predicted. Metabolic maps of eukaryotic metabolism are of higher complexity compared to bacteria since they require information about intracellular compartmentation and intracellular transport (*Duarte, 2004*) and may require multicellular approaches. In plants, aspects of

complex metabolic pathways, such as the energetics of CAM photosynthesis (*Cheung et al., 2014*), and fluxes in C3 and C4 metabolism (*Boyle and Morgan, 2009*; *Gomes de Oliveira Dal'Molin et al., 2011*; *de Oliveira Dal'Molin et al., 2010*; *Arnold and Nikoloski, 2014*; *Saha et al., 2011*) have been elucidated with genome scale models. The C4 cycle is not predicted by these current C4 models unless the C4 cycle is forced by constraints (*Gomes de Oliveira Dal'Molin et al., 2011*; *Mallmann et al., 2014*). In the C4GEM model, the fluxes representing the C4 cycle are a priori constrained to the cell types (*Gomes de Oliveira Dal'Molin et al., 2011*), and in the Mallmann model, the C4 fluxes are induced by activating flux through PEPC (*Mallmann et al., 2014*). Models in which specific a priori constraints activated C4 were successfully used to study metabolism under conditions of photosynthesis, photorespiration, and respiration (*Saha et al., 2011*) and to study N-assimilation under varying conditions (*Simons et al., 2013*). However, they are incapable of testing under which conditions the pathway may evolve.

Schematic models suggest that the C4 cycle evolves from its ancestral metabolic state C3 photosynthesis along a sequence of stages (summarised in *Sage, 2004*; *Bräutigam and Gowik, 2016*). In the presence of tight vein spacing and of photosynthetically active bundle sheath cells (i.e. Kranz anatomy), a key intermediate in which the process of photorespiration is divided between cell types is thought to evolve (*Monson, 1999*; *Sage et al., 2012*; *Heckmann et al., 2013*; *Bauwe, 2010*). The metabolic fluxes in this intermediate suggest an immediate path towards C4 photosynthesis (*Mallmann et al., 2014*; *Bräutigam and Gowik, 2016*). (*Heckmann et al., 2013*) built a kinetic model in which the complex C4 cycle was represented by a single enzyme, PEPC. Assuming carbon assimilation as a proxy for fitness, the model showed that the evolution from a C3 progenitor species with Kranz-type anatomy towards C4 photosynthesis occurs in modular, individually adaptive steps on a Mount Fuji fitness landscape. It is frequently assumed that evolution of C4 photosynthesis requires water limitation (*Bräutigam and Gowik, 2016*; *Heckmann et al., 2013*; *Mallmann et al., 2014*). However, ecophysiological research showed that C4 can likely evolve in wet habitats (*Osborne and Freckleton, 2009*; *Lundgren and Christin, 2017*). CBM presents a possible avenue to study the evolution of C4 photosynthesis including its metabolic complexity *in silico*.

In this study, we establish a generic two-celled, constraint-based model starting from the *Arabidopsis* core model (*Arnold and Nikoloski, 2014*). We test under which conditions and constraints C4 photosynthesis is predicted as the optimal solution. Finally, we test which constraints result in the prediction of the particular C4 modes with their different decarboxylation enzymes. In the process, we demonstrate that evolution is predictable at the molecular level in an eukaryotic system and define the selective pressures and limitations guiding the 'choice' of metabolic flux.

## Results

### The curated *Arabidopsis* core model predicts physiological results

Flux balance analysis requires five types of information, the metabolic map of the organism, the input, the output, a set of constraints (i.e. limitations on input, directionality of reactions, forced flux through reactions), and optimisation criteria for the algorithm which approximate the selective pressures the metabolism evolved under. In this context, inputs define the resources that need to be taken up by the metabolic network to fulfil a particular metabolic function, which is related to the outputs, for example the synthesis of metabolites part of the biomass or other specific products. In CBM, the objective is most likely related to the in- and/or outputs.

For reconstruction of the C3 metabolic map we curated the *Arabidopsis* core model (*Arnold and Nikoloski, 2014*) manually (*Table 1*) to represent the metabolism of a mesophyll cell in a mature photosynthetically active leaf of a C3 plant , further on called *one-cell* model (provided in *Figure 1—source data 1*). The *Arabidopsis* core model is a bottom-up-assembled, large-scale model relying solely on *Arabidopsis*-specific annotations and the inclusion of only manually curated reactions of the primary metabolism. The *Arabidopsis* core model is accurate with respect to mass and energy conservation, allowing optimal nutrient utilisation and biochemically sound predictions (*Arnold and Nikoloski, 2014*).

For the inputs, we considered a photoautotrophic growth scenario with a fixed $CO_2$ uptake of about 20 µmol/(m²s) (*Lacher, 2003*). Light, sulphates, and phosphate are freely available. Due to the observation that nitrate is the main source (80%) of nitrogen in leaves in many species (*Macduff and*

**Table 1.** Curation of the *Arabidopsis* core model from **Arnold and Nikoloski (2014)**.

| Arabidopsis core model | Observation | one-cell model | Reference |
|---|---|---|---|
| NADP-dependent malate dehydrogenases in all compartments | cycles through nitrate reductase to interconvert NAD and NADP | NAD-dependent malate dehydrogenases in all compartments, NADP-dependent malate dehydrogenase only in chloroplast | (*Swarbreck et al., 2008*) |
| Cyclic electron flow | absence of cyclic electron flow | added | (*Shikanai, 2016*) |
| Alternative oxidase | missing alternative routes for electrons to pass the electron transport chain to reduce oxygen | added alternative oxidase reactions to the chloroplast and mitochondria | (*Vishwakarma et al., 2015*) |
| Alanine transferase | No alanine transferase in cytosol Alanine transferase | added | (*Liepman and Olsen, 2003*) |
| Transport chloroplast | no maltose transporter by MEX1 | added | (*Linka and Weber, 2010*) |
| | no glucose transporter by MEX1 and pGlcT MEX1 | added | |
| | no unidirectional transport of ATP, ADP, AMP by BT-like | added | |
| | no Mal/OAA, Mal/Pyr, and Mal/Glu exchange by DiTs | added | |
| | no folate transporter by FBT and FOLT1 | added | |
| Transport Mitochondria | no Mal/OAA, Cit/iCit, Mal/KG exchange by DTC | added | (*Linka and Weber, 2010*) |
| | no H+ importer by UCPs import | added | |
| | no OAA/Pi exchange by DIC1-3 | added | |
| | no ATP/Pi exchange by APCs | added | |
| | no NAD/ADP and NAD/AMP exchange by NDT2 | added | |
| | no ThPP/ATP exchange by TPCs | added | |
| | no Asp/Glu by AGCs | added | |
| | no uncoupled Ala exchange | added | |
| Transport peroxisome | missing NAD/NADH, NAD/ADP, NAD/AMP exchange by PXN | added | (*Linka and Weber, 2010*) |
| | no ATP/ADP and ATP/AMP exchange by PNCs | added | |
| $H^+$ sinks/sources | $H^+$ sinks/source reaction for the cytosol and futile transport cycles introduced by $H^+$-coupled transport reactions | $H^+$ sinks/source reaction added for each compartment | |
| ATPase stoichiometry | False $H^+$/ATP ratios for the plastidal and mitochondrial ATP synthase | $H^+$/ATP ratio set to 3 : 1 (chloroplast) and 4:1 (mitochondria) | (*Petersen et al., 2012*; *Turina et al., 2016*) |
| Alanine/aspartate transferase | no direct conversion of alanine and aspartate | added to cytosol, chloroplast and mitochondria | (*Schultz and Coruzzi, 1995*; *Duff et al., 2012*) |

*Bakken, 2003*), we set nitrate as the sole nitrogen source. If both ammonia and nitrate are allowed, the model will inevitably predict the physiologically incorrect sole use of ammonia since fewer reactions and less energy are required to convert it into glutamate, the universal amino group currency in plants. Water and oxygen can be freely exchanged with the environment in both directions.

To compute the output, we assume a mature fully differentiated and photosynthetically active leaf, which is optimised for the synthesis and export of sucrose and amino acids to the phloem under minimal metabolic effort. Following the examples of models in bacteria, many plant models use a biomass function which assumes that the leaf is required to build itself (*de Oliveira Dal'Molin et al., 2010*; *Arnold and Nikoloski, 2014*; *Saha et al., 2011*) using photoautotrophic that is (*Arnold and Nikoloski, 2014*) or heterotrophic that is (*Cheung et al., 2014*) energy and molecule supply. In plants, however, leaves transition from a sink phase in which they build themselves from metabolites delivered by the phloem to a source phase in which they produce metabolites for other organs

including sink leaves (*Turgeon, 1989*). The composition of *Arabidopsis* phloem exudate (*Wilkinson and Douglas, 2003*) was used to constrain the relative proportions of the 18 amino acids and the ratio of sucrose : total amino acids (2.2 : 1). To account for daily carbon storage as starch for export during the night, we assume that half of the assimilated carbon is stored in the *one-cell* model. We explicitly account for maintenance costs by the use of a generic ATPase and use the measured ATP costs for protein degradation and synthesis of a mature *Arabidopsis* leaf (*Li et al., 2017*) as a constraint. We initially assume a low photorespiratory flux according to the ambient $CO_2$ and $O_2$ partial pressures considering no heat, drought, salt or osmotic stress which may alter the ratio towards higher flux towards the oxygenation reaction.

To develop a largely unconstrained model and detect possible errors in the metabolic map, we initially kept the model unconstrained with regard to fixed fluxes, flux ratios, and reaction directions. Different model iterations were run in (re-)design, simulate, validate cycles against known physiology with errors sequentially eliminated and a minimal set of constraints required for a C3 model recapitulating extant plant metabolism determined. After each change, the CBM predicted all fluxes which were output as a table and manually examined (for example see *Figure 1—source data 2*).

The initial FBA resulted in carbon fixation by enzymes such as the malic enzymes which, in reality, are constrained by the kinetics of the enzymes towards decarboxylation. All decarboxylation reactions were made unidirectional towards decarboxylation to prevent erroneous carbon fixation in the flux distribution. The next iteration of FBA predicted loops through nitrate reductases which ultimately converted NADH to NADPH. We traced this loop to an error in the initial model, in which malate dehydrogenases in the cytosol and mitochondrion were NADP-dependent instead of NAD-dependent. After correction of the co-factor in the *one-cell* model, the loops through nitrate reductases were no longer observed. Another iteration predicted excessive flux through the mitochondrial membrane where multiple metabolites were exchanged and identified missing transport processes as the likely reason. Based on *Linka and Weber (2010)*, we added known fluxes across the mitochondrial and plastidic envelope membranes which remedied the excessive fluxes in the solution. The chloroplastic ADP/ATP carrier protein is constrained to zero flux since its mutant is only affected during the night but not if light is available (*Reiser et al., 2004*).

The obtained flux distribution still contained excessive fluxes through multiple transport proteins across internal membranes which ultimately transferred protons between the organelles and the cytosol. Since for most if not all transport proteins the precise protonation state of metabolites during transport is unknown and hence cannot be correctly integrated into the model, we allowed protons to appear and disappear as needed in all compartments. This provision precludes conclusions about the energetics of membrane transport. ATP generation occurred in a distorted way distributed across different organelles which were traced to the $H^+$ consumption of the ATPases in mitochondria and chloroplasts. The stoichiometry was altered to to 3:1 (chloroplast) and 4:1 (mitochondria) (*Petersen et al., 2012*; *Turina et al., 2016*). We assume no flux for the chloroplastic NADPH dehydrogenase and plastoquinol oxidase because (*Josse et al., 2000*; *Yamamoto et al., 2011*) have shown that their effect on photosynthesis is minor.

In preparation for modelling the C4 cycle, we ensured that all reactions known to occur in C4 (i.e. malate/pyruvate exchange, likely via DiT2 in maize [*Weissmann et al., 2016*], possibly promiscuous amino transferases [*Duff et al., 2012*]) are present in the *one-cell* model, since (*Aubry et al., 2011*) showed that all genes encoding enzymes and transporters underlying the C4 metabolism are already present in the genome of C3 plants. We integrated cyclic electron flow (*Shikanai, 2016*) and alternative oxidases in the mitochondria (*Vishwakarma et al., 2015*), since both have been hypothesised to be important during the evolution and/or execution of the C4 cycle. Models and analysis workflows provided as jupyter notebooks (*Thomas et al., 2016*) are available as supplementary material or can be accessed on GitHub https://github.com/ma-blaetke/CBM_C3_C4_Metabolism (*Blätke, 2019*; copy archived at https://github.com/elifesciences-publications/CBM_C3_C4_Metabolism).

The *one-cell* model comprises in total 413 metabolites and 572 reactions, whereof 139 are internal transporters, 90 are export and eight import reactions (see also below), which are involved in 59 subsystems. *Figure 1* provides an overview of the primary subsystems according to *Arnold and Nikoloski (2014)*.

The *one-cell* model requires a photosynthetic photon flux density (PPFD) of 193.7 μmol/($m^2$s) (*Table 2*). The *one-cell* model takes up the maximal amount of $CO_2$ to produce the maximum

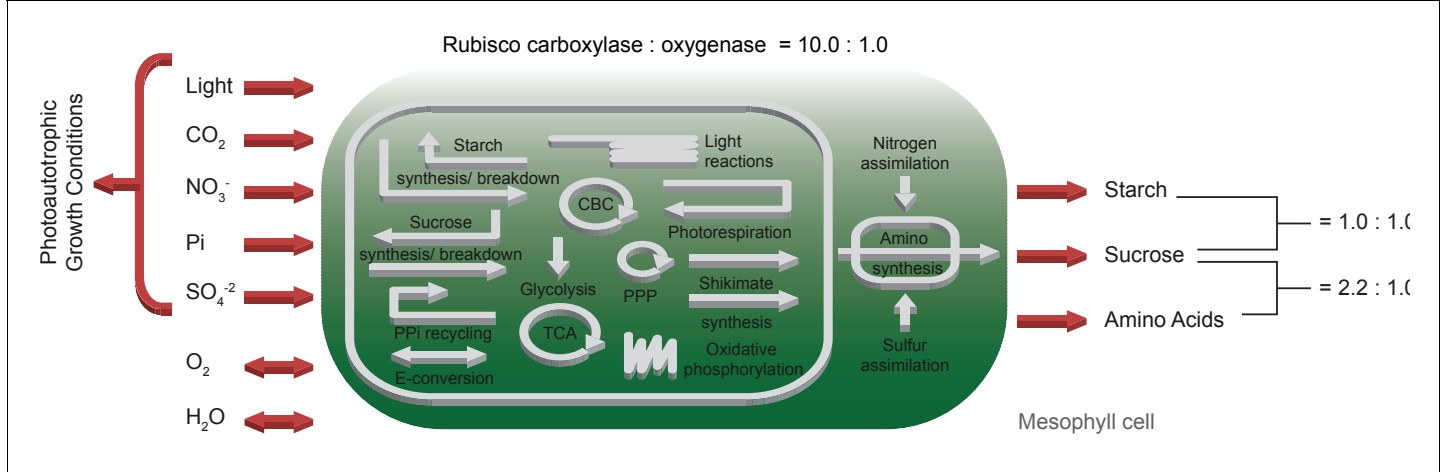

**Figure 1.** Schematic representation of the primary subsystems in the *one-cell* model and the used input/output constraints; adapted from *Arnold and Nikoloski (2014)*.

The online version of this article includes the following source data, source code and figure supplement(s) for figure 1:

**Source data 1.** SBML code of the *one-cell* model.
**Source data 2.** Complete flux solution of the *one-cell* model.
**Source code 1.** Jupyter notebook - Predicted fluxes of C3 metabolism.
**Source code 2.** Jupyter notebook- Effect of the $CO_2$ uptake rate on C3 metabolism.
**Source code 3.** Jupyter notebook - Effect of the PPFD on C3 metabolism.
**Figure supplement 1.** Effect of $CO_2$ and PPFD variation.
**Figure supplement 2.** Energy Flux Distribution in the *one-cell* Model.

amount of phloem sap, as well as 0.8 $\mu$mol/($m^2$s) of $NO_3^-$ and 18.2 $\mu$mol/($m^2$s) of $H_2O$. According to the assumed ratio of sucrose and amino acids in the phloem sap, the flux of sucrose predicted by the model is 0.5 $\mu$mol/($m^2$s) and of amino acids 0.3 $\mu$mol/($m^2$s). The rate of oxygen supply by the network is 20.9 $\mu$mol/($m^2$s). Part of the complete flux table is displayed in *Table 2*; the full table is available, see *Figure 1—source data 2*. The flux table of all reactions did not display circular fluxes, and the reactions were within expected physiological ranges (*Figure 1—source data 2*).

The $CO_2$ uptake rate and the phloem sap output have a positive linear relationship, see *Figure 1—figure supplement 1(A)*. The same is true for the correlation of the PPFD and phloem sap output in the range of 100 $\mu$mol/($m^2$s)–200 $\mu$mol/($m^2$s), see *Figure 1—figure supplement 1(B)*. Above 200 $\mu$mol/($m^2$s), the $CO_2$ uptake rate acts as a limiting factor restricting the increase of phloem sap production. If either the PPFD or the $CO_2$ uptake rate is zero, the phloem sap cannot be

**Table 2.** Input/output fluxes of *one-cell* model in comparison to physiological observations.

| Molecular Species | Flux [$\mu$mol/($m^2$/s)] | Physiological Range [$\mu$mol/($m^2$/s)] | Reference |
|---|---|---|---|
| (i) Inputs | | | |
| Photons | 193.7 | 100 - 400 | *Bailey et al. (2001)* |
| CO2 | 20 | 20 | *Lacher (2003)* |
| $NO_3$ | 0.5 | 0.11 - 0.18 | *Kiba et al. (2012)* |
| $H_2O$ | 18.2 | - | |
| (ii) Outputs | | | |
| $O_2$ | 20.9 | 16.5 | *Sun et al. (1999)* |
| Amino Acids | 0.3 | - | |
| Sucrose/Starch | 0.8 | - | |

Note: $CO_2$ has one carbon per molecule while Sucrose has 12. Starch is configured to have the same number of carbons compared to sucrose while amino acids on average have 5.5 carbons.

produced, compare *Figure 1—figure supplement 1(A) and (B)*. Most of the metabolic processes use ATP/ADP as main energy equivalent (60%), followed by NADP/NADPH (37.5%) and NAD/NADH (2.4%), see *Figure 1—figure supplement 2(D)*. Nearly all ATP is produced by the light reactions (97.2%) and consumed by the reductive pentose phosphate cycle (94.1%), see *Figure 1—figure supplement 2(A)*. The oxidative phosphorylation produces only (1%) of ATP. In proportion, the maintenance cost for protein synthesis and degradation makeup 28% of the respiratory ATP produced by the oxidative phosphorylation (*Figure 1—figure supplement 2(E)*). Similarly, nearly all NADPH is produced by the light reaction (98.9%), which is consumed by the reductive pentose-phosphate cycle (98.3%) as well (*Figure 1—figure supplement 2(B)*). The canonical glycolysis and photorespiration produce nearly equal amounts of NADH, 45% and 47.7%, significantly less NADH is produced through the pyruvate dehydrogenase activity 6.85%. Nitrate assimilation (45%), glutamate biosynthesis (47.7%), glyoxylate cycle (21.6%) and alternative respiration (11.8%) consume the produced NADH (*Figure 1—figure supplement 2(C)*).

## A C4 cycle is predicted under resource limitation

To rebuild the characteristic physiology of C4 leaves, we duplicated the *one-cell* model and connected the two network copies by bi-directional transport of cytosolic metabolites including amino acids, sugars, single phosphorylated sugars, mono-/di-/tri-carboxylic acids, glyceric acids, glycolate, glycerate, glyceraldehyde-3-phosphate, di-hydroxyacetone-phosphate and $CO_2$, see Materials and methods for details. Since CBM is limited to static model analysis, we introduced two Rubisco populations in the bundle sheath network to approximate $CO_2$ concentration-dependent changes in the oxygenation : carboxylation ratio of Rubisco ($v_{RBO}/v_{RBC}$) itself. We kept the native constrained Rubisco population that is forced to undertake oxygenation reactions and added a CCM-dependent Rubisco population which can only carboxylate ribulose 1,5-bisphosphate. The CCM-dependent Rubisco population is only able to use $CO_2$ produced by the bundle sheath network but not environmental $CO_2$ released by the mesophyll. C4 plants have a higher $CO_2$ consumption and thus, an increased $CO_2$ uptake of 40 µmol/(m$^2$s) was allowed (*Leakey et al., 2006*). All other constraints and the objective of the *one-cell* model are maintained in the *two-cell* model, see *Figure 2*.

Initially, we optimised for the classical objective function of minimal total flux through the metabolic network at different levels of photorespiration. These different levels of photorespiration integrate changes to external $CO_2$ concentration and stomatal opening status which is governed by plant water status and biotic interactions. From the complete flux distribution, we extracted fluxes of PEPC and PPDK, the decarboxylation enzymes, Rubisco and metabolite transporter between the two cells to ascertain the presence of a C4 cycle, see *Figure 3* and *Figure 3—figure supplement 1*. At low photorespiratory levels, flux through PEPC is barely detectable (*Figure 3(A)*). If photorespiration increases to moderate levels, flux through PEPC can be predicted and increases to 40 µmol/(m$^2$s), that is all $CO_2$ is funnelled through PEPC, for high photorespiratory fluxes. Concomitant with flux through PEPC, the activity of the decarboxylation enzymes changes (*Figure 3(B)*). At low to intermediate levels of photorespiratory flux, glycine decarboxylase complex activity is predicted to shuttle $CO_2$ to the bundle sheath at up to 4.7 µmol/(m$^2$s). Decarboxylation of C4 acids is initially mostly mediated by PEP-CK and is largely taken over by NADP-ME at high fluxes through photorespiration. Flux through NAD-ME is very low under all photorespiration levels. The decarboxylation enzymes dictate flux through the different Rubiscos in the model (*Figure 3(C)*). At low photorespiratory flux, both the Rubiscos in mesophyll and bundle sheath are active. Only very little flux occurs through the CCM-dependent Rubisco, which is a result of the glycine decarboxylase (*Figure 3(B)*). With increasing photorespiratory flux, this flux through glycine decarboxylase increases (*Figure 3(B)*) and therefore, total Rubisco activity exceeds the carbon intake flux (*Figure 3(C)*). Carbon fixation switches to the CCM-dependent Rubisco with increasing flux through PEPC (*Figure 3(A)*) and the classic C4 cycle decarboxylation enzymes (*Figure 3(B)*). Flux through PPDK mostly reflects flux through PEPC (*Figure 3(D)*). The transport fluxes between the cells change with changing photosynthetic mode (*Figure 3(E and F)*).

At low rates of photorespiration when PEPC is barely active, the only flux towards the bundle sheath is $CO_2$ diffusion (*Figure 3(E)*) with no fluxes towards the mesophyll (*Figure 3(F)*). In the intermediate phase glycolate and glycerate are predicted to be transported and a low-level C4 cycle dependent on the transport of aspartate, malate, PEP and alanine operates (*Figure 3(E) and (F)*). In case of high photorespiratory rates, the exchange between mesophyll and bundle sheath is mainly

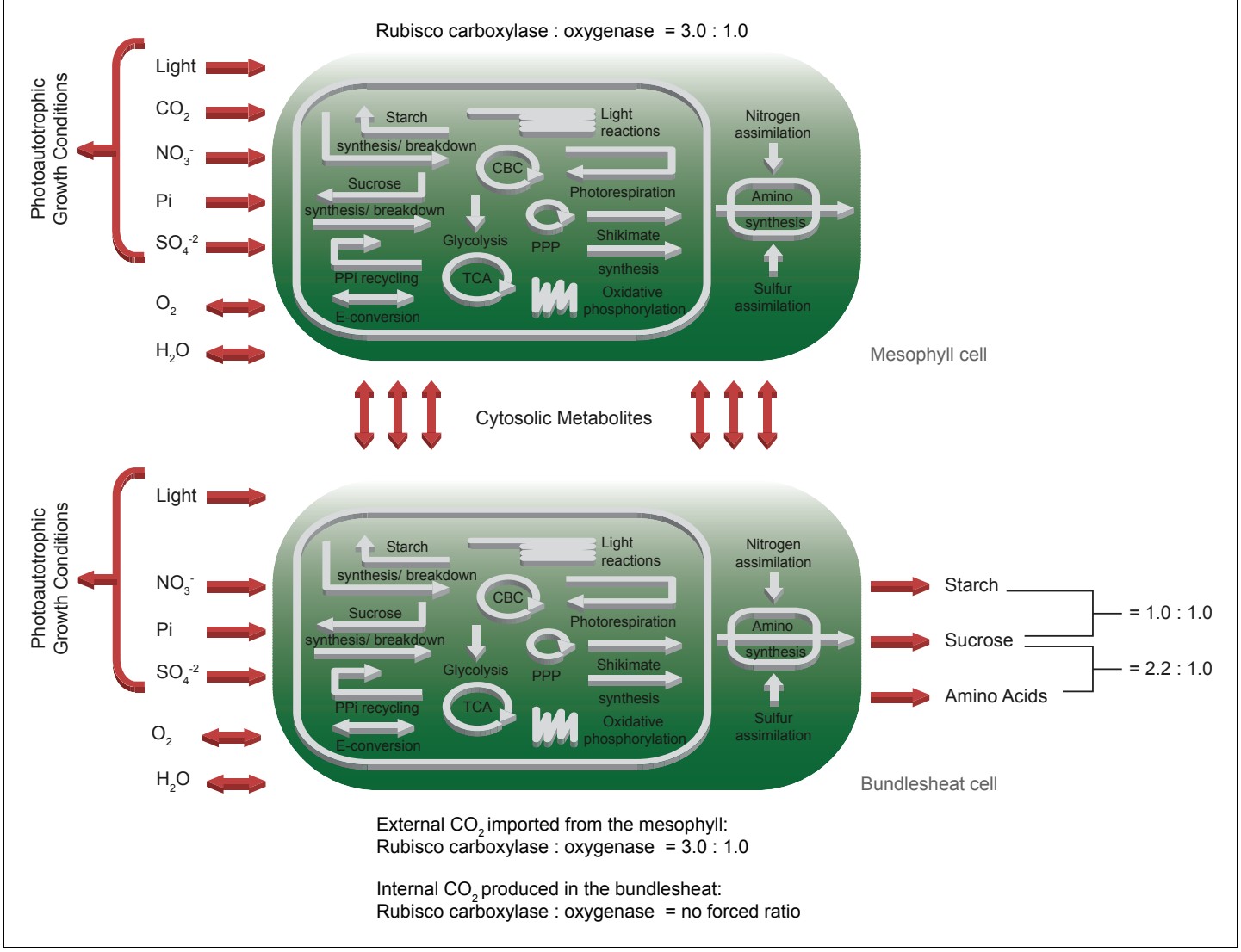

**Figure 2.** Schematic representation of the primary subsystems in the *two-cell* model and the used input/output constraints; adapted from *Arnold and Nikoloski (2014)*.

carried by malate and pyruvate (*Figure 3(E) and (F)*). Flux through PPDK (*Figure 3(D)*) is lower than flux through PEPC (*Figure 3(A)*) at the intermediate stage (*Figure 3(F)*). Evolution of C4 photosynthesis with NADP-ME as the major decarboxylation enzyme is predicted if the photorespiratory flux is high and model optimised for minimal total flux, in other words, resource limitation.

## C4 modes with different decarboxylation enzymes result from different set of constraints

Among the known independent evolutionary events leading to C4 photosynthesis, 20 are towards NAD-ME while 21 occurred towards NADP-ME (*Sage, 2004*). PEP-CK is dominant or at least co-dominant only in *Panicum maximum* (*Bräutigam et al., 2014*), *Alloteropsis semialata semialata* (*Christin et al., 2012*), and in the *Chloridoideae* (*Sage, 2004*). To analyse whether the predicted evolution of the C4 cycle is independent of a particular decarboxylation enzyme, we performed three separate experiments, where only one decarboxylation enzyme can be active at a time. The other decarboxylation enzymes were de-activated by constraining the reaction flux to zero resulting in three different predictions, one for each decarboxylation enzyme. The flux distributions obtained under the assumption of oxygenation : carboxylation ratio of 1 : 3 and minimisation of

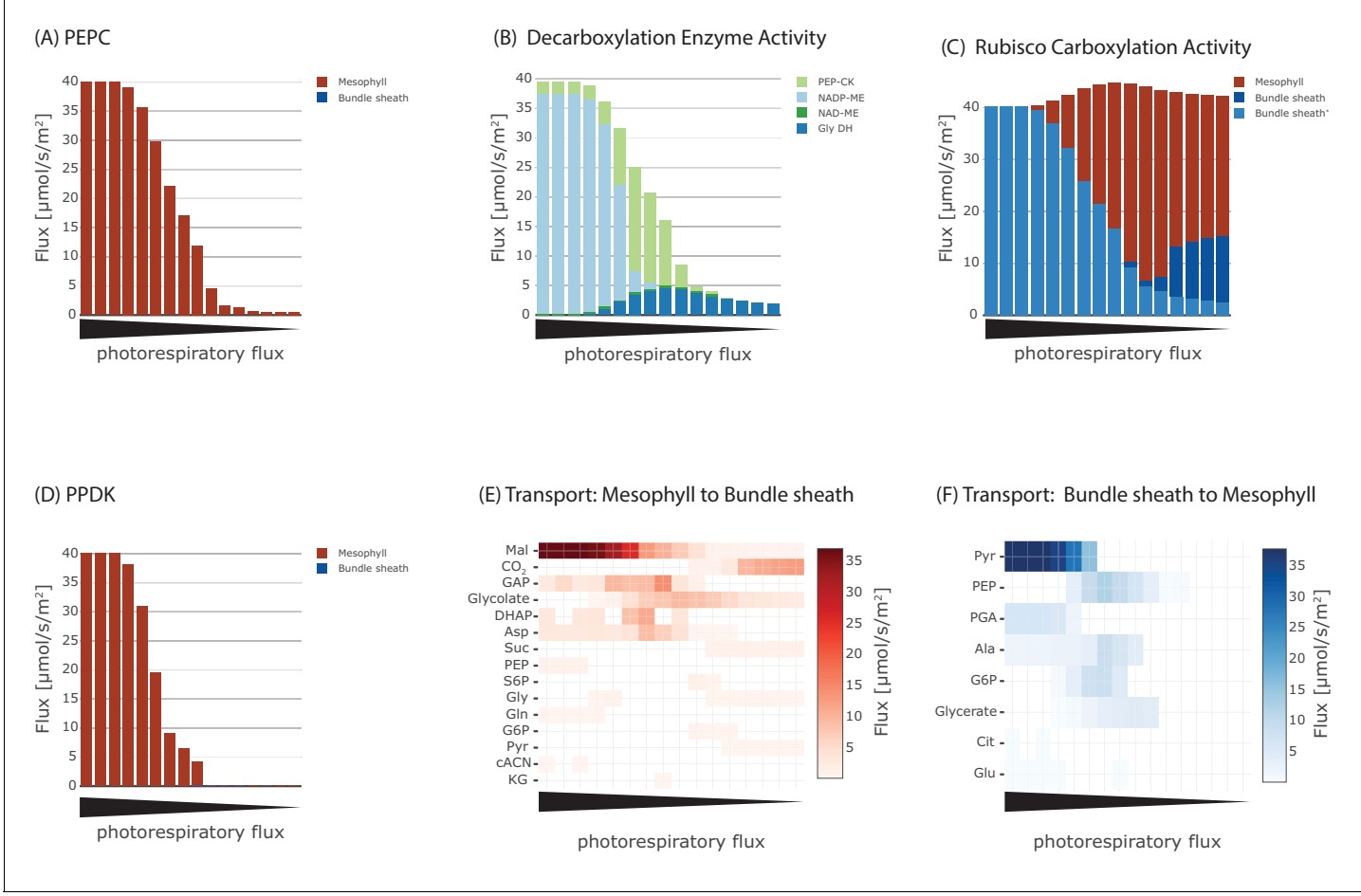

**Figure 3.** Effect of oxygenation : carboxylation ratio on the major steps in C4 cycle, including (**A**) activity of phosphoenolpyruvate carboxylase (PEPC), (**B**) metabolite transport to the bundle sheath, (**C**) activity of Rubisco, (**D**) activity of the decarboxylation enzymes, (**E**) metabolite transport to the mesophyll, and (**F**) activity of pyruvate phosphate dikinase (PPDK).

The online version of this article includes the following source data and figure supplement(s) for figure 3:

**Source code 1.** Jupyter notebook - Analysing the effect of oxygenation : carboxylation ratio on the emergence of the C4 cycle.

**Figure supplement 1.** Flux maps illustrating the effect of the oxygenation : carboxylation ratio of Rubisco on the C3-C4 trajectory.

photorespiration as an additional objective predicts the emergence of a C4 cycle for each known decarboxylation enzyme. To visualise the possible C4 fluxes, the flux distribution for candidate C4 cycle enzymes was extracted from each of the three predictions and visualised as arc width and color (*Figure 4*). While the flux distribution in the mesophyll is identical for three predicted C4 cycles of the decarboxylation enzymes, it is diverse in the bundle sheath due to the different localisation of the decarboxylation and related transport processes, see *Figure 4*. The flux distribution does not completely mimic the variation in transfer acids known from laboratory experiments (*Hatch, 1987*) since all of the decarboxylation enzymes use the malate/pyruvate shuttle. In the case of NAD-ME and PEP-CK, the *two-cell* model also predicts a supplementary flux through the aspartate/alanine shuttle. We tested whether transfer acids other than malate and pyruvate are feasible and explored the near-optimal space. To this end, the model predictions are repeated, allowing deviation from the optimal solution and the changes recorded. Deviations from the optimal solution are visualised as error bars (*Figure 5*). Performing a flux variability analysis (FVA) and allowing the minimal total flux to differ by 1.5%, predicts that for most metabolites which are transferred between mesophyll and bundle sheath, the variability is similar for all three decarboxylation types. For the NAD-ME and PEP-CK types, changes in the near-optimal space were observed for the transfer acids malate, aspartate, pyruvate and alanine. Minor differences were present for triose phosphates and

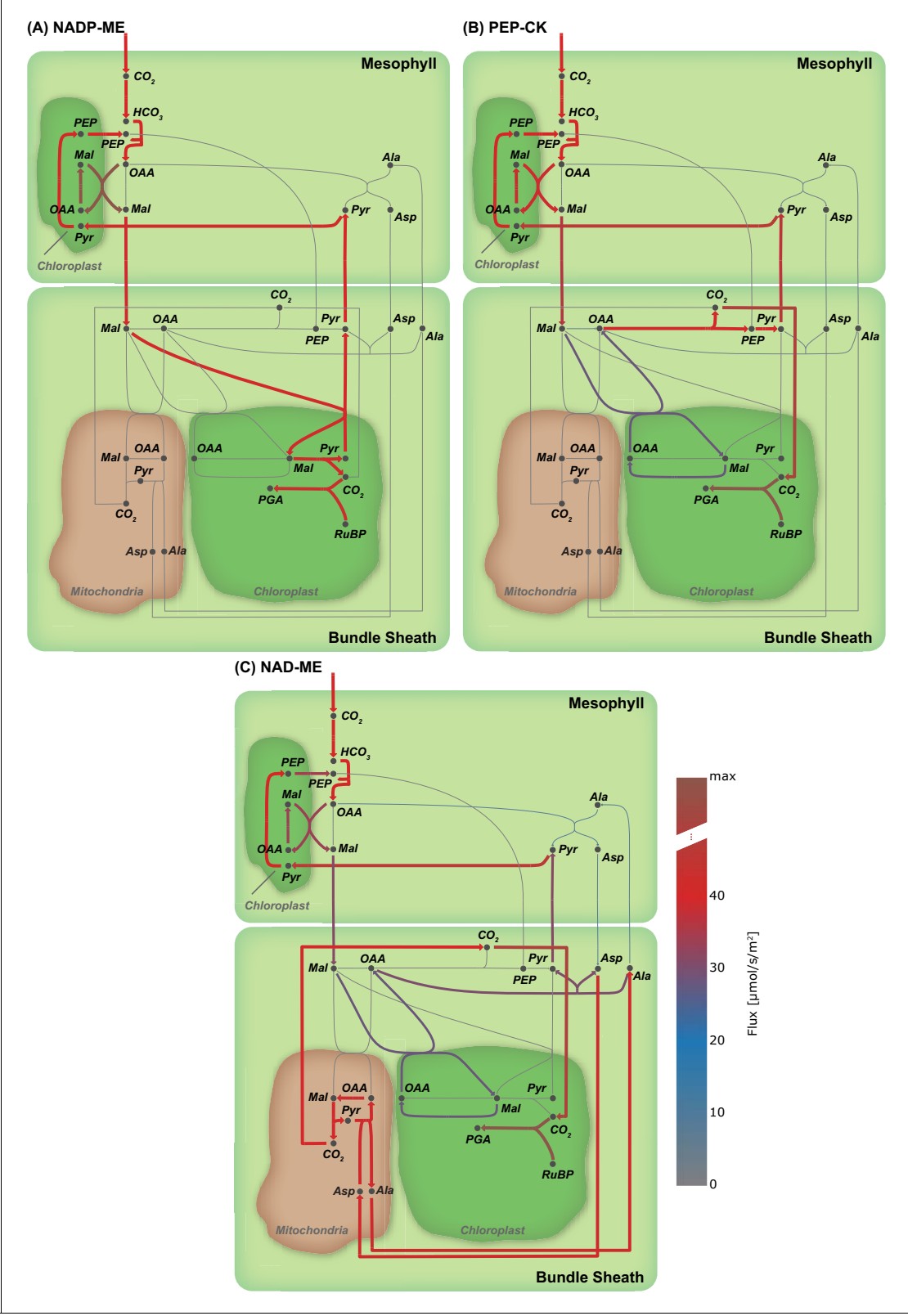

**Figure 4.** Flux maps illustrating the effect of the C4 mode. (**A**) NADP-ME, (**B**) PEP-CK, (**C**) NAD-ME. (Arc width and colour are set relative to flux values in flux, grey arcs - no flux).

The online version of this article includes the following source code for figure 4:

**Source code 1.** Jupyter notebook - Effect of C4 mode on the emergence of the C4 cycle.

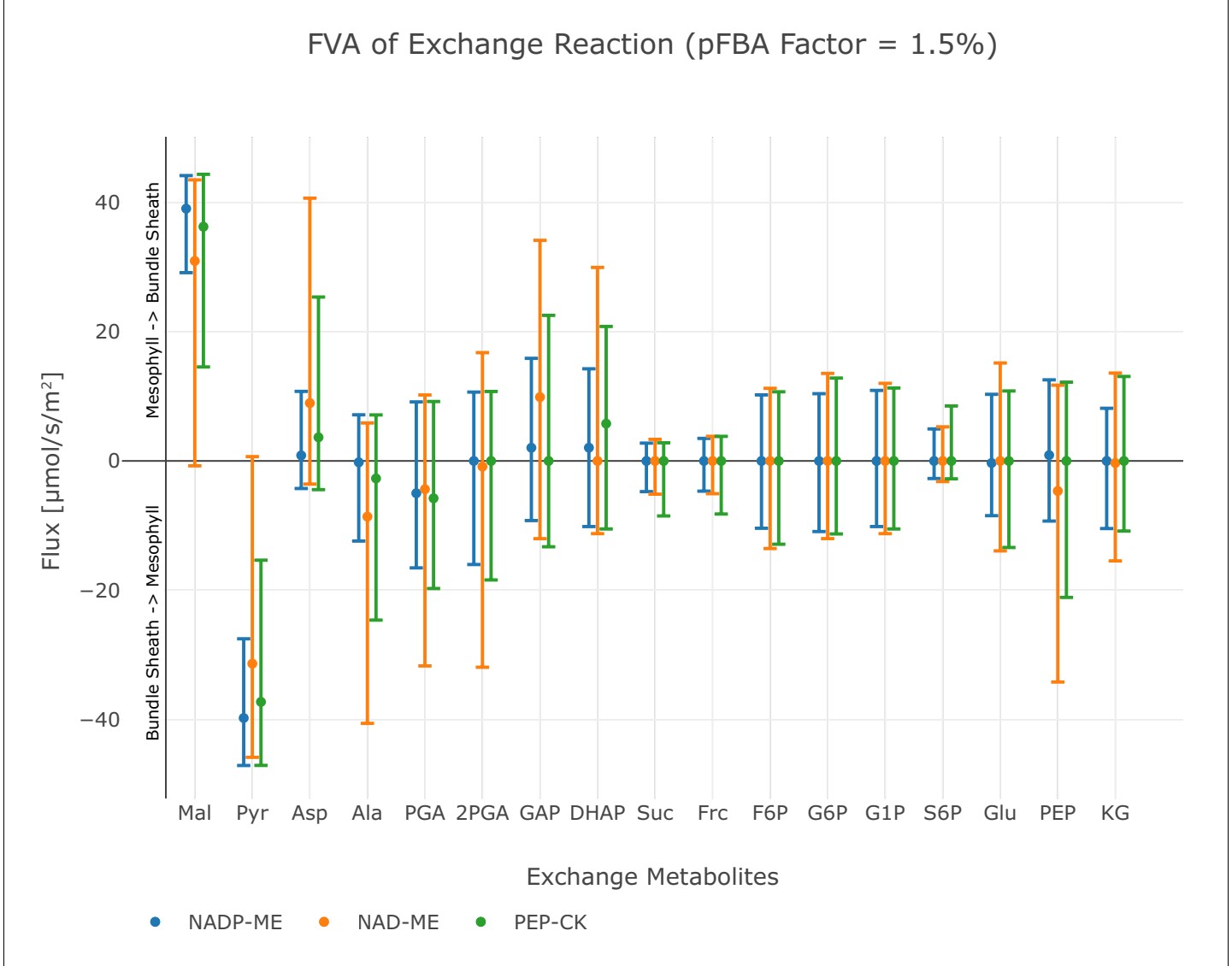

**Figure 5.** Flux variability analysis of metabolite exchange with 1.5% deviation of the total flux minimum. The upper bar defines the maximum exchange flux, while the lower bar defines the minimum exchange flux, points indicate the value of the original flux solution under minimal metabolic effort constraint. Positive flux values correspond to the transport direction from mesophyll to bundle sheath, negative values to the transport direction from bundle sheath to mesophyll, see also *Figure 4—source code 1*.

phosphoglycerates as well as for PEP. For the NADP-ME type, FVA identifies only minor variation (*Figure 5*). In the case of NAD-ME but not in the case of NADP-ME the activity of the malate/pyruvate shuttle can be taken over by the aspartate/alanine shuttle and partly taken over in case of PEP-CK, see *Figure 5*. The aspartate/alanine shuttle is thus only a near-optimal solution when the model and by proxy evolutionary constraints are resource efficiency and minimal photorespiration.

To analyse the effect of other conditions on the particular C4 state, we apply the minimisation of photorespiration as an additional objective to minimal total flux. Since NAD-ME and PEP-CK type plants use amino acids as transfer acids in nature, nitrogen availability has been tagged as a possible evolutionary constraint that selects for decarboxylation by NAD-ME or PEP-CK. When nitrate uptake was limiting, the optimal solution to the model predicted overall reduced flux towards the phloem output (*Figure 6—figure supplement 1*) but reactions were predicted to occur in the same proportions as predicted for unlimited nitrate uptake. Flux through NADP-ME and supplementary flux through PEP-CK dropped proportionally, since restricting nitrogen limits the export of all metabolites from the system and reduced $CO_2$ uptake is observed (*Figure 6—figure supplement 1*).

Similarly, limiting water or $CO_2$ uptake into the model resulted in overall reduced flux towards the phloem output (*Figure 6—figure supplement 1*) but reactions were predicted to occur in the same proportions as predicted for unlimited uptake.

Given that C4 plants sometimes optimise light availability to the bundle sheath (*Bellasio and Lundgren, 2016*) we next explored light availability and light distribution. The model prediction is re-run with changes in the constraints, and the resulting tables of fluxes are queried for $CO_2$ uptake and fluxes through the decarboxylation enzymes. In the experiment, we varied the total PPFD between 0 μmol/(m²s) to 1000 μmol/(m²s) and photon distribution in the range $0.1 \leq PPFD_B / PPFD_M \leq 2$, see *Figure 6*. Under light limitation, if the total PPFD is lower than 400 μmol/(m²s) , the $CO_2$ uptake rate is reduced, leading to a decreased activity of the decarboxylation enzymes (*Figure 6(A)*). PEP-CK is used in the optimal solutions active under light-limiting conditions (*Figure 6*

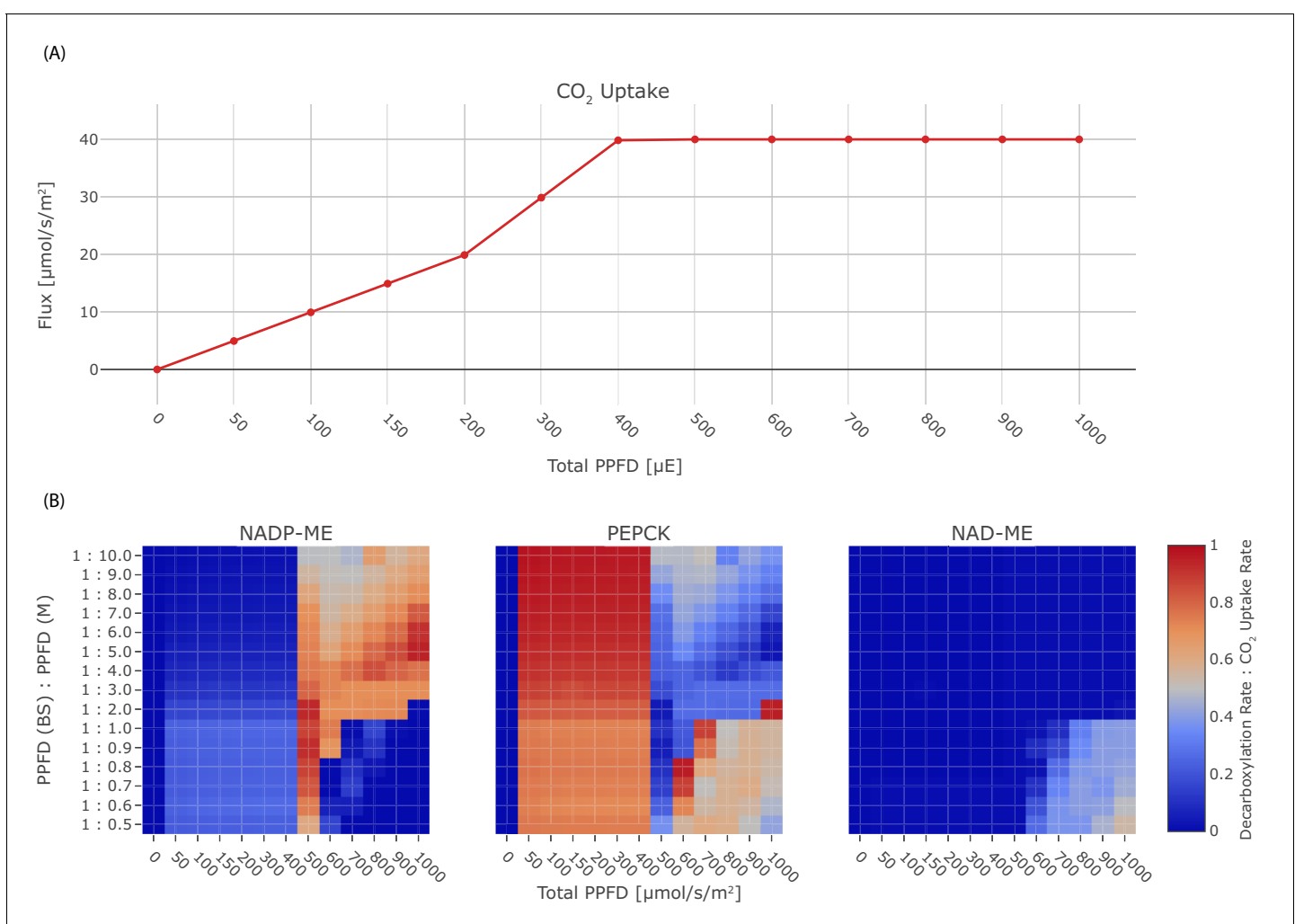

**Figure 6.** Effect of light on the C4 mode. (**A**) $CO_2$ uptake rate in dependence of the total PPFD, (**B**) Heat-maps illustrating the activity of the decarboxylation enzymes PEP-CK, NADP-ME, and NAD-ME relative to the $CO_2$ uptake rate in dependence of the total PPFD and the photon distribution among mesophyll and bundle sheath.

The online version of this article includes the following source data and figure supplement(s) for figure 6:

**Source code 1.** Jupyter notebook - Effect of light on the C4 mode.

**Source code 2.** Jupyter notebook - Effect of $NO_3$ limitation on the C4 mode.

**Source code 3.** Jupyter notebook - Effect of $H_2O$ limitation on the C4 mode.

**Source code 4.** Jupyter notebook - Effect of $CO_2$ limitation on the C4 mode.

**Source code 5.** Jupyter notebook - Effect of malate : aspartate transport ratio on the C4 mode.

**Figure supplement 1.** Effect of other relevant factors on the C4 mode.

*(B)*). Under limiting light conditions, photon distribution with a higher proportion in the bundle sheath shifts decarboxylation towards NADP-ME but only to up to 26%. Under non-limiting conditions, the distribution of light availability determines the optimal decarboxylation enzyme. NADP-ME is the preferred decarboxylation enzyme with supplemental contributions by PEP-CK if light availability is near the threshold of 400 µmol/(m²s) or if at least twice as many photons are absorbed by the mesophyll. Excess light availability and a higher proportion of photons reaching the bundle sheath leads to optimal solutions which favour PEP-CK as the decarboxylation enzyme. In the case of very high light availability and an abrupt shift towards the bundle sheath, NAD-ME becomes the optimal solution (*Figure 6(B)*). NAD-ME is the least favourable enzyme overall, only low activity is predicted under extreme light conditions, where the bundle sheath absorbs equal or more photons than the mesophyll (*Figure 6(B)*). PEP-CK complements the activity of NADP-ME and NAD-ME to 100% in many conditions, meaning the *two-cell* model also predicts the co-existence of PEP-CK/NADP-ME and PEP-CK/NAD-ME mode, while the flux distribution indicates no parallel use of NAD-ME and NADP-ME, compare *Figure 6(B)*.

Finally, we assumed that intercellular transport capacity for charged metabolites might be different between species. Assuming a fixed transport ratio between aspartate and malate (*Figure 6—figure supplement 1D*) introduces a shift in the C4 state. Higher proportions of malate exchange foster the use of NADP-ME (*Figure 6—figure supplement 1D*). In contrast, higher portions of aspartate exchange foster the use of PEP-CK (*Figure 6—figure supplement 1D*).

## Discussion

Evolutionary CBM can suggest the molecular outcomes of past evolutionary events if models are parametrised with objective functions representing possible selective pressures. In the case of C4 photosynthesis, more than sixty independent evolutionary origins represent metabolic types characterised by their decarboxylation enzyme. The selective pressure which drives evolution towards one or the other flux are unknown and were tested using CBM.

### *One-cell* model reflects C3 plant physiology

To analyse evolution towards C4 photosynthesis based on C3 metabolism, a CBM of C3 metabolism is required (*Figure 1*). Design, simulation, validation cycles used current knowledge about plant biochemistry (*Heldt, 2015*) to identify possible errors in the metabolic map required for modelling. Even after error correction (*Table 1*), a significant problem remained, namely excessive fluxes to balance protons in all compartments. This observation leads to the realisation that the biochemical knowledge about transport reactions does not extend to the protonation state of the substrates, which affects all eukaryotic CBM efforts. In plants, predominantly export and vacuolar transport reactions are directly or indirectly coupled with proton gradients to energise transport (*Bush, 1993*; *Neuhaus, 2007*). For chloroplasts and mitochondria, proton-coupled transport reactions have been described but may couple different metabolite transporters together rather than energising them (*Furumoto et al., 2011*). Introducing proton sinks in all compartments solves the immediate modelling problem. However, intracellular transport reactions and their energetic costs are no longer correctly assessed by the model. Despite this band-aid fix which will be required for all eukaryotic constraint-based models which include proton-coupled transport reactions, the curated *one-cell* model correctly predicts energy usage and its distribution (*Figure 1—figure supplement 2* and *Li et al., 2017*). This indicates that in models which exclude vacuolar transport and energised export reactions, energy calculations remain likely within the correct order of magnitude. Overall, our *one-cell* model operates within parameters expected for a C3 plant: The predicted PPFD lies within the range of light intensities used for normal growth condition of *Arabidopsis thaliana*, which varies between 100 µmol/(m²s)–200 µmol/(m²s), see *Table 2*. The gross rate of $O_2$ evolution for a PPFD of 200 µmol/(m²s) is estimated to be 16.5 µmol/(m²s) in the literature (*Sun et al., 1999*), which is in close proximity to the predicted flux of the *one-cell* model, see *Table 2*. For the amount of respiratory ATP that is used for maintenance, (*Li et al., 2017*) predicted an even lower proportion of energy 16%, see *Figure 1—figure supplement 2*. The model's flux map is in accordance with known C3 plant physiology (*Heldt, 2015*), and its input and output parameters match expected values (*Figure 2(B)*). The current model excludes specialised metabolism since the output function focuses solely on substances exported through the phloem in a mature leaf. If the model were to be used to

study biotic interactions in the future, the addition of specialised metabolism in the metabolic map and a new output function would be required.

## The two-cell model predicts a C4 cycle if photorespiration is present

Most evolutionary concepts about C4 photosynthesis assume that selective pressure drives pathway evolution due to photorespiration and carbon limitation (*Heckmann et al., 2013*). Most extant C4 species occupy dry and arid niches (*Edwards et al., 2010*), even more, the period of C4 plant evolution was accompanied with an increased oxygen concentration in the atmosphere (*Sage, 2004*). Therefore, it is frequently assumed that carbon limitation by excessive photorespiration drives the evolution of C4 photosynthesis. Yet, in most habitats plants are limited by nutrients other than carbon (*Agren et al., 2012*; *Körner, 2015*). Ecophysiological analyses also show that C4 can evolve in non-arid habitats (*Liu and Osborne, 2015*; *Lundgren and Christin, 2017*; *Osborne and Freckleton, 2009*). To resolve this apparent contradiction, we tested whether resource limitation may also lead to the evolution of a C4 cycle. We optimised the model approximating resource limitation via an objective function for total minimal flux at different photorespiratory levels. Indeed, with increasing photorespiration, the optimisation for resource efficiency leads to the emergence of the C4 cycle as the optimal solution. Balancing the resource cost of photorespiration against the resource cost of the C4 cycle, the model predicts that N limitation may have facilitated C4 evolution given high levels of photorespiration. Other possible selective pressures such as biotic interactions can currently not be tested using the model since specialised metabolism is not included in the metabolic map or the output function. Extant C4 species have higher C : N ratios reflecting the N-savings the operational C4 cycle enables (*Sage et al., 1987*). The photorespiratory pump using glycine decarboxylase based $CO_2$ enrichment also emerges from the model, showing that C2 photosynthesis is also predicted under simple resource limitation. Indeed N-savings have been reported from C2 plants compared with their C3 sister lineages (*Schlüter et al., 2016a*). Simply minimising photorespiration as the objective function also yields C4 photosynthesis as the optimal solution. Hence, two alternatively or parallelly acting selective pressures towards C4 photosynthesis, limitation in C and/or N, are identified by the model. In both cases, the model correctly predicts the C4 cycle of carboxylation and decarboxylation and the C2 photorespiratory pump as observed in extant plants. The evolution of C4 photosynthesis in response to multiple selective pressures underscores its adaptive value and potential for agriculture. Intermediacy also evolves indicating that it, too, is likely an added value trait which could be pursued by breeding and engineering efforts.

The optimal solutions for the metabolic flux patterns predict an intermediate stage in which $CO_2$ transport via photorespiratory intermediates glycolate and glycerate (*Figure 3(E) and (F)*) and decarboxylation by glycine decarboxylase complex (*Figure 3(B)*) is essential. All of the models of C4 evolution (*Monson, 1999*; *Bauwe, 2010*; *Sage et al., 2012*; *Heckmann et al., 2013*; *Williams et al., 2013*) predict that the establishment of a photorespiratory $CO_2$ pump is an essential intermediate step towards the C4 cycle. The photorespiratory $CO_2$ pump, also known as C2 photosynthesis, relocates the photorespiratory $CO_2$ release to the bundle sheath cells. Plants using the photorespiratory $CO_2$ pump are often termed C3-C4 intermediates owing to their physiological properties (*Sage et al., 2012*). Displaying the flux solution in *Figure 3* on a metabolic map in *Figure 3—figure supplement 1* clearly illustrates that increasing photorespiratory flux through Rubisco drives the two-cell metabolic model from C3 to C4 metabolism by passing the C3-C4 intermediate state. On the C3-C4 trajectory, the activity of Rubisco is shifted from the mesophyll to the bundle sheath, as well as from the constrained to the CCM-dependent Rubisco population as a consequence of the increased costs of photorespiration under increased $p_{O_2} : p_{CO_2}$ ratio, see *Equation 5*. The increase of the oxygenation rate in the photorespiration constraint drives the reprogramming of the metabolism to avoid oxygenation by establishing the C4 cycle. Therefore, our analysis recovers the evolutionary C3-C4 trajectory and confirms the emergence of a photorespiratory $CO_2$ pump as an essential step during the C4 evolution also under optimisation for resources (*Heckmann et al., 2013*). The model may also provide a reason for why some plant species have halted their evolution in this intermediary phase (*Scheben et al., 2017*). Under the conditions of resource limitations and intermediate photorespiration, the model predicts intermediacy as the optimal solution. In a very narrow corridor of conditions, no further changes are required to reach optimality and the model thus predicts that a small number of species may remain intermediate.

## *Two-cell* model realises different C4 states

Since the model predicts C4 metabolism without specific constraints, different input and reaction constraints can be tested for their influence on the molecular nature of the C4 cycle. This approach may identify the selective pressure and boundaries limiting evolution. Initial optimisation without additional constraints or input limitations predict a C4 cycle based on decarboxylation by NADP-ME (*Figure 3* and *Figure 3—figure supplement 1(A)*). This prediction recapitulates intuition; the NADP-ME based C4 cycle is considered the 'most straight forward' incarnation of C4 photosynthesis, it is always explained first in textbooks and is a major focus of research. The NADP-ME based cycle thus represents the stoichiometrically optimal solution when resource limitation or photorespiration are considered. Once NADP-ME is no longer available via constraint, PEP-CK and NAD-ME become optimal solutions albeit with a prediction of malate and pyruvate as the transfer acids (*Figure 6*). The FVA identified aspartate and alanine as slightly less optimal solutions (*Figure 5*). Since *in vivo* this slightly less optimal solution has evolved in all NAD-ME origins tested to date, kinetic rather than stoichiometric reasons suggest themselves for the use of aspartate and alanine (*Bräutigam et al., 2018*).

## Light is a potential evolutionary driver for the different C4 states

Since all extant C3 species and therefore also the ancestors of all C4 species contain all decarboxylation enzymes (*Aubry et al., 2011*), it is unlikely that unavailability of an enzyme is the reason for the evolution of different decarboxylation enzymes in different origins (*Sage, 2004*). Stochastic processes during evolution, that is up-regulation of particular enzyme concentrations via changes in expression and therefore elements *cis* to the gene (*Bräutigam and Gowik, 2016*), may have played a role in determining which C4 cycle evolved. Alternatively, environmental determinants may have contributed to the evolution of different C4 cycles. Physiological experiments have pointed to a connection between nitrogen use efficiency and type of decarboxylation enzyme (*Pinto et al., 2016*). Hence the variation in nitrogen input to the model was tested for their influence on optimal solutions with regard to decarboxylation enzymes. Input limitation of nitrogen, water as a metabolite, and $CO_2$ limited the output of the system but did not change the optimal solution concerning decarboxylation *Figure 6—figure supplement 1* making it an unlikely candidate as the cause. Differences in nitrogen use is possibly a consequence of decarboxylation type.

In some grasses, light penetrable cells overlay the vascular bundle leading to different light availability (summarised in *Bellasio and Lundgren, 2016* and *Karabourniotis et al., 2000*) and hence light availability and distribution were tested (*Figure 6(B)*). Changes in light input and distribution of light input between mesophyll and bundle sheath indeed altered the optimal solutions (*Figure 6(B)*). The changes in the solution can be traced to the energy status of the plant cells. For very high light intensities, the alternative oxidases in the mitochondria are used to dissipate the energy and hence a path towards NAD-ME is paved. Under light limitation, the C4 cycle requires high efficiency and hence PEP-CK which, at least in part allows energy conservation by using PEP rather than pyruvate as the returning C4 acid, is favoured. Interestingly, the sensitivity of different species towards environmental changes in light is influenced by the decarboxylation enzyme present (*Sonawane et al., 2018*). NADP-ME species are less compromised compared to NAD-ME species by shade possibly reflecting an evolutionary remnant as NAD-ME is predicted to emerge only in high light conditions. PEP-CK is more energy efficient compared to malic enzyme based decarboxylation which requires PEP recycling by PPDK at the cost of two molecules of ATP (*Figure 3(D)*). Notably, two C4 plants known to rely on PEP-CK *P. maximum* and *A. semialata* (African accessions) are shade plants which grow in the understory (*Lundgren and Christin, 2017*). PEP-CK can be co-active with NADP-ME and NAD-ME (*Figure 6(B)*). This co-use of PEP-CK with a malic enzyme has been shown in C4 plants (*Pick et al., 2011*; *Wingler et al., 1999*) and explained as an adaptation to different energy availability and changes in light conditions (*Pick et al., 2011*; *Bellasio and Griffiths, 2014*). Dominant use of PEP-CK in the absence of malic enzyme activity as suggested (*Figure 3(B)*, *Figure 3—figure supplement 1* and *Figure 4*) is rare *in vivo* (*Ueno and Sentoku, 2006*) but observed in *P. maximum* and in *A. semialata*. While the model predictions are in line with ecological observations, we cannot exclude that kinetic constraints (i.e. [*Bräutigam et al., 2018*]) may also explain why a stoichiometrically optimal solution such as the NADP-ME cycle is not favoured in nature where NADP-ME and NAD-ME species evolve in nearly equal proportions (*Sage, 2004*).

## Conclusion

CBM of photosynthetically active plant cells revealed a major knowledge gap impeding CBM, namely the unknown protonation state of most transport substrates during intracellular transport processes. When photoautotrophic metabolism was optimised in a single cell for minimal metabolic flux and therefore, optimal resource use, C3 photosynthetic metabolism was predicted as the optimal solution. Under low photorespiratory conditions, a two-celled model which contains a CCM-dependent Rubisco optimised for resource use, still predicts C3 photosynthesis. However, under medium to high photorespiratory conditions, a molecularly correct C4 cycle emerged as the optimal solution under resource limitation and photorespiration reduction as objective functions which points to resource limitation as an additional driver of C4 evolution. Light and light distribution was the environmental variable governing the choice of decarboxylation enzymes. Modelling compartmented eukaryotic cells correctly predicts the evolutionary trajectories leading to extant C4 photosynthetic plant species.

# Materials and methods

## Flux Balance Analysis

Flux balance analysis (FBA) is a CBM approach (*Orth et al., 2010*) to investigate the steady-state behaviour of a metabolic network defined by its stoichiometric matrix $S$. By employing linear programming, FBA allows computing an optimised flux distribution that minimises and/or maximises the synthesis and/or consumption rate of one specific metabolite or a combination of various metabolites. Next to the steady-state assumption and stoichiometric matrix $S$, FBA relies on the definition of the reaction directionality and reversibility, denoted by the lower bound $v_{min}$ and upper bound $v_{max}$ , as well as the definition of an objective function $z$. The objective function $z$ defines a flux distribution $v$, with respect to an objective $c$.

$$\min/\max \quad z_{FBA} = c^T v$$
$$\text{s.t.}$$
$$S \cdot v = 0$$
$$v_{min} \leq v \leq v_{max}$$

(1)

The degeneracy problem, the possible existence of alternate optimal solutions, is one of the major issues of constraint-based optimisation, such as FBA (*Mahadevan and Schilling, 2003*). To avoid this problem, we use the parsimonious version of FBA (pFBA) (*Lewis et al., 2010*). This approach incorporates the flux parsimony as a constraint to find the solution with the minimum absolute flux value among the alternative optima, which is in agreement with the assumption that the cell is evolutionary optimised to allocate a minimum amount of resources to achieve its objective.

$$\min/\max \quad z_{pFBA} = \sum |v_i|$$
$$\text{s.t.}$$
$$S \cdot v = 0$$
$$v_{min} \leq v \leq v_{max}$$
$$c^T v = z_{FBA}$$

(2)

All FBA experiments in this study employ pFBA and are performed using the cobrapy module in a python 2.7 environment run on a personal computer (macOS Sierra, 4 GHz Intel Core i7, 32 GB 1867 MHz DDR3). All FBA experiments are available as jupyter notebooks in the supplementary material and can also be accessed and executed from the GitHub repository https://github.com/mablaetke/CBM_C3_C4_Metabolism (*Blätke, 2019*; copy archived at https://github.com/elifesciences-publications/CBM_C3_C4_Metabolism).

## Generic model for C3 metabolism

### Metabolic model

The generic model representing the metabolism of a mesophyll cell of a mature photosynthetically active C3 leaf, further on called *one-cell* model, is based on the *Arabidopsis* core model

(*Arnold and Nikoloski, 2014*). The model is compartmentalised into cytosol (c), chloroplast (h), mitochondria (m), and peroxisome (p). Each reaction in the *Arabidopsis* core model (*Arnold and Nikoloski, 2014*) was compared with the corresponding entry in AraCyc (*Mueller et al., 2003*). Based on the given information, we corrected co-factors, gene associations, enzyme commission numbers and reversibility (information from BRENDA [*Schomburg et al., 2002*] were included). The gene associations and their GO terms (*Ashburner et al., 2000*) of the cellular components were used to correct the location of reactions. Major additions to the model are the cyclic electron flow (*Shikanai, 2016*), alternative oxidases in mitochondria and chloroplast (*Vishwakarma et al., 2015*), as well as several transport processes between the compartments and the cytosol (*Linka and Weber, 2010*). NAD-dependent dehydrogenase to oxidise malate is present in all compartments (*Gietl, 1992*; *Berkemeyer et al., 1998*), which excludes the interconversion of NAD and NADP by cycles through the nitrate reductase present in the *Arabidopsis* core model. Correctly defining the protonation state of the metabolites in the various cellular compartments is a general drawback of metabolic models due to the lack of knowledge in that area. This issue mainly affects biochemical reactions and transport reactions involving protons. We added a sink/source reaction for protons in the form:

$$\leftrightarrow H\_\{x\} \quad x = c, h, m, p \tag{3}$$

to all compartments to prevent futile fluxes of protons and other metabolites coupled through the proton transport. The curated *one-cell* model is provided in *Figure 1—source data 1*.

## Import

As in *Arnold and Nikoloski (2014)*, we assume photoautotrophic growth conditions. Only the import of light, water, $CO_2$, inorganic phosphate (Pi), nitrate/ammonium, and sulphates/hydrogen sulphide is allowed, compare *Table 3*. More specifically, we do only allow for nitrate uptake, since it is the main source (80%) of nitrogen in leaves (*Macduff and Bakken, 2003*). The $CO_2$ uptake is limited to 20 μmol/(m$^2$s) (*Lacher, 2003*). Therefore, the carbon input constrains the model.

## Export

In contrast to *Arnold and Nikoloski (2014)*, we focus on mature, fully differentiated and photosynthetic active leaves supporting the growth of the plant through the export of nutrients in the phloem sap, mainly sucrose and amino acids. An output reaction for sucrose *Ex_Suc* is already included in the model. An additional export reaction *Ex_AA* represents the relative proportion of 18 amino acids

**Table 3.** Flux boundary constraints of Im-/export reactions

| Input (Reaction ID) | Flux [μmol/(m$^2$s)] | |
| --- | --- | --- |
| | Lower bound | Upper bound |
| Photons (Im_hnu) | 0 | inf |
| C0$_2$ (Im_CO2) | 0 | 20 |
| NO$_3$ (Im_NO3) | 0 | inf |
| NH$_4^+$ (Im_NH4) | 0 | 0 |
| SO$_4^{2-}$ (Im_SO4) | 0 | inf |
| H$_2$S (Im_H2S) | 0 | inf |
| Pi | 0 | inf |
| H$_2$O (Im_H2O) | -inf | inf |
| O$_2$ (Im_O2) | -inf | inf |
| Amino Acids (Ex_AA) | 0 | inf |
| Surcose (Ex_Suc) | 0 | inf |
| Starch (Ex_starch) | 0 | inf |
| Other export reactions | 0 | 0 |

-inf/inf is approximated by $-10^6$ / $10^6$

**Table 4.** Maintenance costs by compartment

| Compartment | Flux [μmol/(m²s)] |
| --- | --- |
| cytosol | 0.0427 |
| chloroplast | 0.1527 |
| mitochondria | 0.0091 |
| peroxisome | 0.0076 |

in the phloem sap of *Arabidopsis* as stoichiometric coefficients in accordance to experimentally measured data from *Wilkinson and Douglas (2003)*. The ratio of exported sucrose : total amino acid is estimated to be 2.2 : 1 (*Wilkinson and Douglas, 2003*). This ratio is included as a flux ratio constraint of the reactions *Ex_Suc* and *Ex_AA*. Furthermore, it is known that the export of sucrose and the formation of starch is approximately the same (*Stitt and Zeeman, 2012*), which is reflected by the flux ratio constraint $v_{Ex\_Suc} : v_{Ex\_starch}$ = 1 : 1. The model allows for the export of water and oxygen. The flux of all other export reactions is set to 0, see *Table 3* for a summary.

## Additional Constraints

We explicitly include the maintenance costs in our model to cover the amounts of ATP that is used to degradation and re-synthesis proteins for each compartment. (*Li et al., 2017*) specifies the ATP costs for protein degradation and synthesis of each compartment of a mature *Arabidopsis* leaf. Based on the given data, we were able to calculate the flux rates to constrain the maintenance reactions in each compartment (*Table 4*).

The *one-cell* model contains maintenance reactions only for the cytsol (*NGAM_c*), chloroplast (*NGAM_h*) and mitochondria (*NGAM_m*) in the form:

$$ATP\_\{x\} + H2O\_\{x\} \rightarrow ADP\_\{x\} + H\_\{x\} + Pi\_\{x\} \quad x = c, h, m \tag{4}$$

An equivalent maintenance reaction cannot be formulated for the peroxisome since in the *one-cell* model ATP/ADP are not included as peroxisomal metabolites. The flux through the maintenance reactions is fixed to the determined maintenance costs given in *Table 4*. The peroxisomal maintenance costs are added to the cytosolic maintenance costs.

The $CO_2$ and $O_2$ partial pressures determine the ratio of the oxygenation : carboxylation rate of Rubisco (given by reactions *RBO_h* and *RBC_h*) and can be described by the mathematical expression:

$$\frac{v_{RBO\_h}}{v_{RBC\_h}} = \frac{1}{S_R} \cdot \frac{p_{O_2}}{p_{CO_2}}, \tag{5}$$

where $S_R$ specifies the ability of Rubisco to bind $CO_2$ over $O_2$. In the case of a mature leave and ambient $CO_2$ and $O_2$ partial pressures in temperate regions with adequate water supply, the ratio $v_{RBO_h}/v_{RBC_h}$ is fixed and is predicted to be 10%, which is encoded by an additional flux ratio constraint.

We assume no flux for the chloroplastic NADPH dehydrogenase (*iCitDHNADP_h*) and plastoquinol oxidase (*AOX4_h*) because (*Josse et al., 2000*) and (*Yamamoto et al., 2011*) have shown that their effect on the photosynthesis is minor.

## Objective

In accordance with the assumption of mature, fully differentiated and photosynthetic active leaf, the model's objective is to maximise the phloem sap output defined by reactions *Ex_Suc* and *Ex_AA*. Additionally, we assume that the involved plant cells put only a minimal metabolic effort, in the form of energy and resources, into the production of phloem sap as possible. This assumption is in correspondence with minimising the nitrogen investment by reducing the number of enzymes that are active in a metabolic network. Therefore, we perform a parsimonious FBA to minimise the total flux.

For enhanced compliance with the recent standards of the systems biology community, the *one-cell* model is encoded in SBML level 3. Meta-information on subsystems, publications, cross-

references are provided as evidence code in the form of MIRIAM URI's. FBA related information, gene association rules, charge and formula of a species element are encoded using the Flux Balance Constraints package developed for SBML level 3. All fluxes in the model are consistently defined as $\mu mol/(m^2 s)$.

## Generic model for C4 metabolism

### Metabolic model

The generic model of C4 metabolism, short *two-cell* model, comprises two copies of the *one-cell* model to represent one mesophyll and one bundle sheath cell. Reactions and metabolites belonging to the metabolic network of the mesophyll are indicated with the prefix *[M]*, whereas the prefix for the bundle sheath is *[B]*. The separate mesophyll and bundle sheath networks are connected via reversible transport reactions of the cytosolic metabolites indicated with the prefix *[MB]*, *Figure 2*. The C4 evolution not only confined Rubisco to the bundle sheath cells, the $CO_2$ concentrating mechanism steadily supplies Rubisco with $CO_2$ in such a way that the oxygenation rate is negligible. Therefore, the bundle sheath network is equipped with two Rubisco populations. The native Rubisco population binds external $CO_2$ and adheres to forced oxygenation : carboxylation ratios, where the optimised evolutionary population binds only internal $CO_2$ and the carboxylation occurs independently of the oxygenation. External $CO_2$ is defined as *[B]_CO2_ex_{c,h}* supplied by the mesophyll network. Internal $CO_2$ given by *[B]_CO2_{c,h,m}* originates from reactions in the bundle sheath network producing $CO_2$. External $CO_2$ in the bundle sheath network is only allowed to move to the chloroplast *[B]_Tr_CO2h_Ex* and to react with Rubisco *[B]_RBC_h_Ex*. The differentiation of two Rubisco populations binding either external or internal $CO_2$ approximates the concentration-dependent shift of the oxygenation : carboxylation ratio.

### Imports

As for the *one-cell* model, we assume photoautotrophic growth conditions, see *Table 3*. During C4 evolution the $CO_2$ assimilation became more efficient allowing higher $CO_2$ assimilation rates. *Zea mays* achieves up to 40 $\mu mol/(m^2 s)$ (*[M]_Im_CO2*) (*Rozema, 1993*). We assume that the $CO_2$ uptake from the environment by the bundle sheath has to be bridged by the mesophyll. Therefore, the input flux of [B]_Im_CO2 is set to zero.

### Exports

The outputs of the *one-cell* model are transferred to the mesophyll and bundle sheath network, as well as the corresponding flux ratios, see *Table 3*.

### Additional Constraints

The ATP costs for cell maintenance in the *genC3* model are assigned to both cell types in the *two-cell* model. Due to declining $CO_2$ concentrations over evolutionary time and/or adverse conditions which close the stromata, the oxygenation : carboxylation ratio of the native Rubisco population in the bundle sheath and the mesophyll is increased and can be predicted as 1 : 3, the corresponding flux ratios are adapted accordingly. Furthermore, we assume that the total photon uptake in the mesophyll and bundle sheath is in the range of 0 $\mu mol/(m^2 s)$ to 1000 $\mu mol/(m^2 s)$. Since they are more central in the leaf, the photon uptake by the bundle sheath must be equal or less compared to the mesophyll. The mesophyll and bundle sheath networks are connected by a range of cytosolic transport metabolites including amino acids, sugars (glucose, fructose, sucrose, trehalose, ribose), single phosphorylated sugar (glucose-6-phosphate, glucose-1-phosphate, fructose-6-phosphate, sucrose-6-phosphate), mono-/di-/tri-carboxylic acids (phosphoenolpyruvate, pyruvate, citrate, cis-aconitate, isocitrate, $\alpha$-ketoglutarate, succinate, fumarate, malate), glyceric acids (2-Phosphoglycerate, 3-Phosphoglycerate), glycolate, glycerate, glyceraldehyde-3-phosphate, di-hydroxyacetone-phosphate and $CO_2$. Nucleotides, NAD/NADH, NADP/NADPH, pyrophosphate, inorganic phosphate are not considered as transport metabolites. Oxaloacetate has been excluded as transport metabolite since concentrations of oxaloacetate are very low *in vivo* and it is reasonably unstable in aqueous solutions. Other small molecules that can be imported by the bundle sheath from the environment, as well as protons and $HCO_3$, are not exchanged between the two cell types.

## Objective
The maximisation of the phloem sap output through the bundle sheath and the minimisation of the metabolic effort are kept as objectives in the *two-cell* model.

## Acknowledgements
We like to thank Udo Gowik (Carl von Ossietzky University Oldenburg, Germany) and Urte Schlüter (Heinrich-Heine-University Düsseldorf, Germany) for critically revising this manuscript.

## Additional information

### Funding
The authors declare that there was no funding for this work.

### Author contributions
Mary-Ann Blätke, Resources, Data curation, Software, Formal analysis, Investigation, Visualization, Methodology, Writing—original draft; Andrea Bräutigam, Conceptualization, Supervision, Validation, Writing—original draft

### Author ORCIDs
Mary-Ann Blätke (iD) https://orcid.org/0000-0002-4790-7377

### Decision letter and Author response
Decision letter https://doi.org/10.7554/eLife.49305.sa1
Author response https://doi.org/10.7554/eLife.49305.sa2

## Additional files

### Supplementary files
• Transparent reporting form

### Data availability
All data generated or analysed during this study are included in the manuscript and supporting files. We provide jupyter notebooks as documentation for all the in silico experiments using constraint-based modelling and additional python code for Figure 1, 3, 4, 6, as well as the metabolic network used as source data for Figure 1 which can be accessed and executed from the GitHub repository https://github.com/ma-blaetke/CBM_C3_C4_Metabolism (copy archived at https://github.com/elifesciences-publications/CBM_C3_C4_Metabolism).

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
