## [Decision Letter]

**Acceptance summary:**

The environmental and evolutionary forces pushing the transition between different types of photosynthesis in plants has been a question of long-standing interest. In this work, the authors utilized computational modelling of plant metabolism in conjunction with the environmental parameters to test this question. They identify a surprisingly limited set environmental parameters and metabolic features that are sufficient to explain how these transitions occur and why.

**Decision letter after peer review:**

Thank you for submitting your article "Evolution of C4 photosynthesis predicted by constraint-based modelling" for consideration by *eLife*. Your article has been reviewed by three peer reviewers, one of whom is a member of our Board of Reviewing Editors, and the evaluation has been overseen by Christian Hardtke as the Senior Editor. The following individuals involved in review of your submission have agreed to reveal their identity: Gavin Conant (Reviewer #2); Amy Marshall-Colon (Reviewer #3).

The reviewers have discussed the reviews with one another and the Reviewing Editor has drafted this decision to help you prepare a revised submission.

The main requests for clarity by the reviewers revolve around these three general areas.

1) Commenting on how this model may or may not fit to reality. This includes issues of assumptions inherent to the CBM model, flux issues in *Arabidopsis* that are lineage specific, and how existing physiological measures could be used to support the modeling.

2) Noted issues with the Jupyter Notebook.

3) Making the writing more accessible in how the data shows the claims and what are hypothesis generated by the models that need testing.

*Reviewer #1:*

The authors use a flux based optimization approach to investigate the C3/C4 transitions and the different C4 possibilities. Overall, this is a very interesting manuscript. As an editorial note, as a general biologist, I could understand the main conclusions due to the explicit statements making these conclusions. It would help however to walk the general biologists through the results and figures more to help them understand how the data makes these conclusions. I was able to divine this from the data but it did take effort that a general biologist may not put into the work.

I do have one point of clarification. The model is built on the *Arabidopsis* flux model but I was unsure if this is the generic flux model or the one that incorporates specialized metabolism. It might make some sense to comment on how using the *Arabidopsis* C3 map as a resource from the flux model given that *Arabidopsis* has very different sulfate/sulfur fluxes in comparison to non-Brassicaceous plants. This links directly to sulfur-amino acids and glutathione metabolism due to the heavy requirement for glucosinolate metabolism as shown by the work of Conant, Pires and colleagues. At the very least some discussion about how large carbon sinks not in the model may or may not change the conclusions would be helpful especially to guide future modelling efforts.

*Reviewer #2:*

In this manuscript, the authors develop a constraint-based metabolic model (CBM) of C4 photosynthesis using known C3 metabolic models with additional constraints. They compare the C3 and C4 models to see if various evolutionary pressures can drive a metabolic shift from C3 to C4 photosynthesis. They find that selection for minimizing metabolic turnover will drive metabolism total the C4 solution with differing light conditions controlling the appearance of different classes of C4.

I enjoyed reading this manuscript, and I commend the authors on their approach to an important and complex topic. However, I think that the manuscript could use improvement in two key areas.

First, while the authors discuss at several points how their models are consistent with physiological data, these comparisons appear to be vague and qualitative. For instance, in the ninth paragraph of the subsection “The curated *Arabidopsis* core model predicts physiological results”, the authors give many numerical predictions of their model, but no comparative values from real plants. Obviously, many model parameters are not measurable (hence the value of the model). But if the model is reproducing physiology, the authors should clearly specify how and with what level of error. Note also that there are advanced approaches for fitting models such as these to real data on gene expression levels, trying to make models that more closely follow observed physiology (1). The authors certainly need not apply them, but the text (c.f., especially the Abstract) as written rather implies that CBMs have not been successfully used with eukaryotes, which is not true. (Also note that "eukaryote" is not synonymous with "multicellular" (2)).

My second concern is that the authors do not make enough of an effort to guide the reader through the potential confounds of CBM for problems such as this one. For instance, in the last paragraph of the subsection “C4 modes with different decarboxylation enzymes result from different set of constraints”, the authors reject nitrogen-limitation as a driver of C4 evolution. However, if compounds that are composed of a good deal of nitrogen are used as catalysts in a CBM, the model will not predict a need for high levels of nitrogen because those compounds are regenerated. I believe that the various C4 carbon shuttles function in exactly this way, and a CBM that seeks only to synthesize sugars and amino acids and has no biomass component will not predict an increased nitrogen requirement for C4 even if that requirement exists, because the compounds requiring that nitrogen cycle in the model. To be clear, I suspect the authors are correct that light is a more likely driver of C4 evolution than nitrogen limitation. But I am not convinced the CBM proves this.

Likewise, CBMs "overpredict" solutions to metabolic problems because they are not kinetic. Hence, the fact that the base 2-cell model does not differentiate between the C4 classes does not mean that there are not kinetic differences that are evolutionarily important.

References:

1) Shlomi T, Cabili MN, Herrgard MJ, Palsson BO, and Ruppin E (2008) Network-based prediction of human tissue-specific metabolism. Nature Biotechnology 26(9):1003-1010.

2) Duarte NC, Herrgård MJ, and Palsson BØ (2004) Reconstruction and validation of *Saccharomyces cerevisiae* iND750, a fully compartmentalized genome-scale metabolic model. Genome Research 14:1298-1309.

*Reviewer #3:*

The manuscript by Blätke and Brautigam provides a novel method using constraint based modeling to predict the selective pressures that resulted in C4 metabolism. The take-away from this study is that C4 metabolism emerged from increasing photorespiration with concomitant resource efficiency, and that light is a driver for different C4 states. The model presented here overcomes some of the limitations of previous flux models of C3 and C4 metabolism, and makes an important contribution to the field. However, the main take-aways of this study are not easy for the reader to pull out as they are buried within the manuscript. This very interesting manuscript could be improved by streamlining the text and more clearly summarizing the main results either in the section titles or as a couple of sentences at the end of each Results section. Much of the Discussion is redundant to the results, so can be shortened for clarity. Likewise, the Discussion would be more interesting if the authors took a deeper dive into the potential outcomes and applicability of their results; how can the information revealed in this study guide future efforts for engineering C4 metabolism into C3 crops? Are there potential transcriptional regulatory mechanisms that could be investigated to evolve C4 metabolism in C3 crops? It is also unclear if the authors altered exogenous CO_2_ concentration in the model; this should either be more clearly explained how it was done and the results, or why it wasn't done. Finally, it is wonderful that the authors include their Jupyter Notebooks and source data for the reader; however, we identified a few issues that made it difficult to run the code. First, there is no Readme file to follow.

Second, the code would not run with the current installations of the Cobra and Escher packages; this was resolved by changing cobra.io.sbml3.read_sbml_model() to cobra.io.sbml.read_sbml_model() in the load_sbml_model.py file as cobra.io.sbml3 does not appear to exist in the current version of the cobra package. The escher package returned a syntax error from the plots.py file in the initialization step of the jupyter notebook files. In order to run the files, we commented out (removed) any uses of the escher package in the code.

[Editors' note: further revisions were requested prior to acceptance, as described below.]

Thank you for resubmitting your work entitled "Evolution of C4 photosynthesis predicted by constraint-based modelling" for further consideration by *eLife*. Your revised article has been reviewed by three peer reviewers, including Dan Kliebenstein as the Reviewing Editor, and the evaluation has been overseen by Christian Hardtke as the Senior Editor.

The manuscript has been improved but there are some remaining issues that need to be addressed before final acceptance, as outlined below.

There are a few suggestions where in a couple of places the writing could be tightened to improve the accessibility for the general reader.

*Reviewer #1:*

The authors have nicely addressed my previous concerns and I have nothing further to add.

*Reviewer #2:*

In this manuscript, the authors develop a constraint-based metabolic model (CBM) of C4 photosynthesis using known C3 metabolic models with additional constraints. They compare the C3 and C4 models to see if various evolutionary pressures can drive a metabolic shift from C3 to C4 photosynthesis.

The authors have addressed most of my concerns, but I still find the description of the effects of nitrogen, water and CO_2_ limitations confusing. The authors write "the optimal solution to the model predicted the same behavior" and "resulted in the same optimal solution as unlimited uptake with the differences of proportionally lower flux overall."

A quick reading of these statements makes one believe that the optimal solution is unchanged with N, H_2_O or CO_2_ limitation, which is common with FBA when a metabolite is *not* truly limiting. I think the authors' point is actually that the proportional fluxes are the same but with different magnitudes, but neither the text nor Figure 6 nor Figure 6—figure supplement 1 seem to me to be crystal clear on this point.

Introduction, second paragraph: "prohibitive" is a slightly odd word to use here. Lower speeds are very likely maladaptive, but are not lethal for growth.

Subsection “The curated *Arabidopsis* core model predicts physiological results”, first paragraph: I would add a few words on what these inputs and outputs are.

*Reviewer #3:*

The authors addressed each of our comments to various extents. We appreciate the clear headings that provide the reader with the main findings of each section. However, we still find the summary at the end of each section to be lacking. That said, the authors did a good job of streamlining the section on the C4 cycle under resource limitation to reveal the key insights. This helped to reduce the redundancy of the Discussion section. Importantly, the authors corrected the Jupyter Notebooks issue such that now they can easily be run. The conclusion paragraph still needs work to achieve clarity and a logical flow of ideas. For example, the second sentence seems out of place. Otherwise we are satisfied with the revised version.

---

## [Author Response]

Reviewer #1:The authors use a flux based optimization approach to investigate the C3/C4 transitions and the different C4 possibilities. Overall, this is a very interesting manuscript. As an editorial note, as a general biologist, I could understand the main conclusions due to the explicit statements making these conclusions.

Reviewer 3 asked for streamlining the manuscript, and indeed, some sections between the Results and Discussion were repetitive. We have excised those Discussion elements from the Results and hope that the clarity has not suffered.

It would help however to walk the general biologists through the results and figures more to help them understand how the data makes these conclusions. I was able to divine this from the data but it did take effort that a general biologist may not put into the work.

Thank you for this comment. We have now added an introductory sentence for each figure that explains how the visualization is generated and why. In essence, all figures and tables are partial visualizations of the flux distribution predicted by the model under different constraints.

I do have one point of clarification. The model is built on the Arabidopsis flux model but I was unsure if this is the generic flux model or the one that incorporates specialized metabolism. It might make some sense to comment on how using the Arabidopsis C3 map as a resource from the flux model given that Arabidopsis has very different sulfate/sulfur fluxes in comparison to non-Brassicaceous plants. This links directly to sulfur-amino acids and glutathione metabolism due to the heavy requirement for glucosinolate metabolism as shown by the work of Conant, Pires and colleagues.

The *Arabidopsis* core model from Arnold and Nikoloski, 2014, describes only the primary metabolism. We added two sentences to make explicit the absence of specialized/secondary metabolism. Our curation process based on the *Arabidopsis* core model did not add reactions part of the specialized metabolism. The duplicated two-cell model we are using to investigate C4 also contains no additional specialized metabolic routes; only transport reactions have been added.

At the very least some discussion about how large carbon sinks not in the model may or may not change the conclusions would be helpful especially to guide future modelling efforts.

We agree that the current model is unable to integrate biotic interactions since it lacks specialized metabolism in both metabolic map and output function. We have inserted one sentence each in the discussion of the results of the single-cell model and the two-cell model to make this absence explicit for the reader.

Reviewer #2:[…] I enjoyed reading this manuscript, and I commend the authors on their approach to an important and complex topic. However, I think that the manuscript could use improvement in two key areas.First, while the authors discuss at several points how their models are consistent with physiological data, these comparisons appear to be vague and qualitative. For instance, in the ninth paragraph of the subsection “The curated Arabidopsis core model predicts physiological results”, the authors give many numerical predictions of their model, but no comparative values from real plants. Obviously, many model parameters are not measurable (hence the value of the model). But if the model is reproducing physiology, the authors should clearly specify how and with what level of error.

Thank you for making this point. The original plant model on which the single-cell model was based contained multiple reactions which lead to flux distributions through enzymes and transporters not known to carry large fluxes in plants. This included circular fluxes across membranes for simple transport processes since transport reactions were missing. All of these were corrected, leading to a state more in agreement with the physiological state and the biochemistry according to current textbooks.

To our knowledge, MFA in plant leaves is currently limited to the Calvin Benson Bassham cycle (https://www.pnas.org/content/pnas/111/47/16967.full.pdf), and hence it is presently not possible for us to benchmark the model predictions against a known flux distribution.

Hence, we limit our comparison to output from the model and make the comparison explicit in a table in Figure 1B, which includes the model predictions and the measured values. A more detailed interpretation and comparison to experimental measurements can be found in the Discussion section, for the particular case “One cell model reflects C3 plant physiology”. We now also include a flux distribution table. Using this table, one can check the fluxes against one’s own expectations of plant metabolism without having to re-run the model.

Note also that there are advanced approaches for fitting models such as these to real data on gene expression levels, trying to make models that more closely follow observed physiology (1). The authors certainly need not apply them, but the text (c.f., especially the Abstract) as written rather implies that CBMs have not been successfully used with eukaryotes, which is not true. (Also note that "eukaryote" is not synonymous with "multicellular" (2)).

We did not attend to leave the impression that CBM has not already been successfully applied to eukaryotic/multicellular systems. We instead aimed to say that we see potential in the applications of CBMs when it comes to systems more complex than single-celled bacteria. According to the comment of reviewer 3, we have rewritten the Abstract and rephrased the respective part in the new Abstract and in the Introduction.

My second concern is that the authors do not make enough of an effort to guide the reader through the potential confounds of CBM for problems such as this one. For instance, in the last paragraph of the subsection “C4 modes with different decarboxylation enzymes result from different set of constraints”, the authors reject nitrogen-limitation as a driver of C4 evolution. However, if compounds that are composed of a good deal of nitrogen are used as catalysts in a CBM, the model will not predict a need for high levels of nitrogen because those compounds are regenerated. I believe that the various C4 carbon shuttles function in exactly this way, and a CBM that seeks only to synthesize sugars and amino acids and has no biomass component will not predict an increased nitrogen requirement for C4 even if that requirement exists, because the compounds requiring that nitrogen cycle in the model. To be clear, I suspect the authors are correct that light is a more likely driver of C4 evolution than nitrogen limitation. But I am not convinced the CBM proves this.

We obviously were not clear enough in our writing. Indeed, the model predicts C4 evolution if we optimize for minimal total flux. To us, this indicates that C4 is predicted to evolve under conditions where resources (i.e. N) are limited. The C:N ratios of C4 plants and intermediates support this prediction.

Light and its distribution “only” influences which decarboxylation path is favoured, while limited N uptake does not influence the decarboxylation path. Reviewer 3 criticized that the manuscript was not written clearly enough. The Results section was edited to clarify this issue in the text.

Likewise, CBMs "overpredict" solutions to metabolic problems because they are not kinetic. Hence, the fact that the base 2-cell model does not differentiate between the C4 classes does not mean that there are not kinetic differences that are evolutionarily important.

We agree that kinetic constraints are likely also critical. To avoid the impression that CBM provides a final solution, we have inserted a sentence in the Discussion, which points out the critical role of kinetic constraints.

References:1) Shlomi T, Cabili MN, Herrgard MJ, Palsson BO, and Ruppin E (2008) Network-based prediction of human tissue-specific metabolism. Nature Biotechnology 26(9):1003-1010.2) Duarte NC, Herrgård MJ, and Palsson BØ (2004) Reconstruction and validation of Saccharomyces cerevisiae iND750, a fully compartmentalized genome-scale metabolic model. Genome Research 14:1298-1309.Reviewer #3:The manuscript by Blätke and Brautigam provides a novel method using constraint based modeling to predict the selective pressures that resulted in C4 metabolism. The take-away from this study is that C4 metabolism emerged from increasing photorespiration with concomitant resource efficiency, and that light is a driver for different C4 states. The model presented here overcomes some of the limitations of previous flux models of C3 and C4 metabolism, and makes an important contribution to the field. However, the main take-aways of this study are not easy for the reader to pull out as they are buried within the manuscript. This very interesting manuscript could be improved by streamlining the text and more clearly summarizing the main results either in the section titles or as a couple of sentences at the end of each Results section.

We have removed the intermittent partial discussion in favour of single sentences at the end of each section. Section headers were also changed.

Much of the Discussion is redundant to the Results, so can be shortened for clarity.

Thank you for pointing this out. We have removed the intermittent discussion from the Results and limited interpretation in the Results to single sentences at the end of each section.

Likewise, the Discussion would be more interesting if the authors took a deeper dive into the potential outcomes and applicability of their results; how can the information revealed in this study guide future efforts for engineering C4 metabolism into C3 crops? Are there potential transcriptional regulatory mechanisms that could be investigated to evolve C4 metabolism in C3 crops?

Thank you for pointing out this oversight. We have now inserted into the Discussion an evaluation of C4 and intermediacy as breeding and engineering targets for agriculture.

With regard to the transcriptional regulation, we think CBM cannot reveal potential transcriptional regulatory mechanisms. In Mallmann et al., 2014, and in the review Bräutigam and Gowik, 2016, we point out that evolution most likely occurs in *cis* in the C4 cycle genes; in Kühlahoglu et al. we have proposed that the C4 genes are likely hooked into the photosynthetic regulon.

It is also unclear if the authors altered exogenous CO_2_ concentration in the model; this should either be more clearly explained how it was done and the results, or why it wasn't done.

The CO_2_ concentration is reflected in changes to photorespiration. There are multiple factors (i.e. external CO_2_ concentration and stomatal opening, which in turn is influenced by plant water status and biotic interactions) which alter CO_2_ availability in the plant. Hence we opted to converge them in altered photorespiratory flux. We now added a sentence to make this fact explicit.

Finally, it is wonderful that the authors include their Jupyter Notebooks and source data for the reader; however, we identified a few issues that made it difficult to run the code. First, there is no Readme file to follow.Second, the code would not run with the current installations of the Cobra and Escher packages; this was resolved by changing cobra.io.sbml3.read_sbml_model() to cobra.io.sbml.read_sbml_model() in the load_sbml_model.py file as cobra.io.sbml3 does not appear to exist in the current version of the cobra package. The escher package returned a syntax error from the plots.py file in the initialization step of the jupyter notebook files. In order to run the files, we commented out (removed) any uses of the escher package in the code.

Thank you for putting in the effort to run the code and point out the deficiencies in the documentation. We updated the file “READM.md” in the GitHub repository to link listed figure supplements in chronological order to the corresponding jupyter notebooks. In addition, we added a file “environment.yml” which contains detailed information about the used module versions to avoid any version conflicts. With the addition of the “environment.yml” the jupyter notebooks can now easily be executed at “https://mybinder.org/”, links are provided in the “READM.md”. Please be aware, that we provide the GitHub repository as zipped folder, but it is more convenient to view and access files in the GitHub repository itself:

https://github.com/ma-blaetke/CBM_C3_C4_Metabolism

We added a short comment about the GitHub Repository to the end of the first Results section and to the “Materials and methods”.

[Editors' note: further revisions were requested prior to acceptance, as described below.]

Reviewer #2:[…] The authors have addressed most of my concerns, but I still find the description of the effects of nitrogen, water and CO_2_ limitations confusing. The authors write "the optimal solution to the model predicted the same behavior" and "resulted in the same optimal solution as unlimited uptake with the differences of proportionally lower flux overall."A quick reading of these statements makes one believe that the optimal solution is unchanged with N, H_2_O or CO_2_ limitation, which is common with FBA when a metabolite is not truly limiting. I think the authors' point is actually that the proportional fluxes are the same but with different magnitudes, but neither the text nor Figure 6 nor Figure 6—figure supplement 1 seem to me to be crystal clear on this point.We have edited the text passage to reflect our results better and now clearly state that the solutions result in reduced flux but proportionally identical solutions. We have also edited the figure legend of the figure in question to clarify the point further.Introduction, second paragraph: "prohibitive" is a slightly odd word to use here. Lower speeds are very likely maladaptive, but are not lethal for growth.Subsection “The curated Arabidopsis core model predicts physiological results”, first paragraph: I would add a few words on what these inputs and outputs are.

We also corrected the typos and wording as suggested by the reviewers. As suggested by reviewer #2, we also added a short explanation on inputs and outs after introducing the basic set up of flux balance analysis.

Reviewer #3:The authors addressed each of our comments to various extents. We appreciate the clear headings that provide the reader with the main findings of each section. However, we still find the summary at the end of each section to be lacking. That said, the authors did a good job of streamlining the section on the C4 cycle under resource limitation to reveal the key insights. This helped to reduce the redundancy of the Discussion section. Importantly, the authors corrected the Jupyter Notebooks issue such that now they can easily be run. The conclusion paragraph still needs work to achieve clarity and a logical flow of ideas. For example, the second sentence seems out of place. Otherwise we are satisfied with the revised version.

The concern of reviewer #3 with the conclusion statement was addressed by rephrasing to improve the flow.